# EVALUATING VISUAL COUNTERFACTUAL EXPLAINERS

## ABSTRACT

Explainability methods have been widely used to provide insight into the decisions made by statistical models, thus facilitating their adoption in various domains within the industry. Counterfactual explanation methods aim to improve our understanding of a model by perturbing samples in a way that would alter its response in an unexpected manner. This information is helpful for users and for machine learning practitioners to understand and improve their models. Given the value provided by counterfactual explanations, there is a growing interest in the research community to investigate and propose new methods. However, we identify two issues that could hinder the progress in this field. (1) Existing metrics do not accurately reflect the value of an explainability method for the users. (2) Comparisons between methods are usually performed with datasets like CelebA, where images are annotated with attributes that do not fully describe them and with subjective attributes such as "Attractive". In this work, we address these problems by proposing an evaluation method with a principled metric to evaluate and compare different counterfactual explanation methods. The evaluation is based on a synthetic dataset where images are fully described by their annotated attributes. As a result, we are able to perform a fair comparison of multiple explainability methods in the recent literature, obtaining insights about their performance. We make the code and data public to the research community.

## 1 INTRODUCTION

The popularity of deep learning methods is a testament to their effectiveness across a multitude of tasks in different domains. This effectiveness has led to their widespread industrial adoption (e.g., self-driving cars, screening systems, healthcare, *etc.*), where the need to explain a model's decision becomes paramount. However, due to the high level of complexity of deep learning models, it is difficult to understand their decision making process (Burkart & Huber, 2021). This ambiguity has slowed down the adoption of these systems in critical domains. Hence, in order to ensure algorithmic fairness in deep learning and to identify potential biases in training data and models, it is key to explore the reasoning behind their decisions (Buhrmester et al., 2021).

In an attempt to convincingly tackle the *why* question, there has been a surge of work in the field of explainability for machine learning models (Joshi et al., 2018; Mothilal et al., 2020; Rodríguez et al., 2021). The goal of this field is to provide explanations for the decisions of a classifier, which often come in the form of counterfactuals. These conterfactuals provide insight as to why the output of the algorithms is not any different and how it could be changed (Goyal et al., 2019). Basically a counterfactual explanation answers the question: *"For situation X why was the outcome Y and not Z"*, describing what changes in a situation would have produced a different decision.

Ideally, counterfactual methods produce explanations that are interpretable by humans while reflecting the factors that influence the decisions of a model (Mothilal et al., 2020). So given an input sample and a model, a counterfactual explainer would perturb certain attributes of the sample, producing a counterexample *i.e., counterfactual*, that shifts the model's prediction, thus revealing which semantic attributes the model is sensitive to. In this work, we focus on the image domain given the recent surge of explainability methods for image classifiers (Joshi et al., 2018; Rodríguez et al., 2021; Singla et al., 2019; Chang et al., 2018). A particular challenge of the image domain is that changes in the pixel space are difficult to interpret and resemble adversarial attacks (Goodfellow et al., 2014b) (see Figure 4 for an example), so current explainers tend to search for counterfactuals in a latent space produced

by, e.g., a variational autoencoder (VAE) (Kingma & Welling, 2013), or by conditioning on annotated attributes (Denton et al., 2019; Joshi et al., 2018; Singla et al., 2019; Rodríguez et al., 2021).

Although explanations produced in the latent space are easier to interpret than in the pixel space, they depend on the chosen or learned decomposition of the input into attributes or latent factors. In the case of VAEs, these factors could be misaligned with the real underlying generating process of the images. Moreover, different methods in the literature rely on different autoencoding architectures or generative models to infer semantic attributes from images, which make their counterfactual search algorithms not comparable. In the case of annotations, since datasets do not provide access to the whole data generating process, they tend to focus on arbitrary aspects of the input (such as facial attributes for CelebA (Liu et al., 2015b)), ignoring other aspects that could influence a classifier's decision boundaries such as illumination, background color, shadows, etc. This raises the need for evaluating explainers on a known set of attributes that represent the real generative factors of the input. In this work, we propose to fill this gap by introducing a new explainability benchmark based on a synthetic image dataset, where we model the whole data generating process and samples are fully described by a controlled set of attributes.

An additional challenge when evaluating explainers is that there is no consensus on the metric that should be used. While there has been some effort to provide a general metric to evaluate explainers (Mothilal et al., 2020; Rodríguez et al., 2021), most of the proposed metrics could be easily gamed to maximize the score of a given explainer without actually improving its quality for a user. For example, since current metrics reward producing many explanations, the score can be increased by (i) producing random samples that cannot be related to the ones being explained. This has motivated measuring the *proximity* of explanations (Mothilal et al., 2020). (ii) Repeating the same explanation many times. This motivates measuring *diversity* (Mothilal et al., 2020). However, we found that it is possible to maximize existing diversity measures by always performing the same perturbation to a sensitive attribute while performing random perturbations to the rest of attributes that describe a sample. As a result, although one counterfactual changing the sensitive attribute would suffice, an explainer could obtain a higher score by producing more redundant explanations. We argue that instead of providing many explanations, explainers should be designed to produce the minimal set of counterfactuals that represent each of the factors that influence a model's decision. (iii) Providing uninformative or trivial explanations (Rodríguez et al., 2021). This has motivated us to compare the model's predictions with those expected from an "oracle classifier". Model predictions that deviate from the expected value are more informative than those that behave as expected (see A.2. In this work, we address these problems by proposing a fair way to evaluate and compare different counterfactual explanation methods.

cierOur contributions can be summarized as follows: (i) we present a benchmark to evaluate counterfactuals generated by any explainer in a fair way (Section 3); (ii) we offer insights on why existing explainability methods have strong limitations such as an ill-defined oracle (Section 3.3); (iii) we introduce a new set of metrics to evaluate the quality of counterfactuals (Section 3.4); and (iv) we evaluate 6 explainers across different dataset configurations (Section 4).

## 2 RELATED WORK

**Explainability methods.** Since most successful machine learning models are uninterpretable (He et al., 2016; Jégou et al., 2017; LeCun et al., 1989), modern explainability methods have emerged to provide explanations for these types of models, which are known as post-hoc methods. An important approach to post-hoc explanations is to establish feature importance for a given prediction. These methods (Guidotti et al., 2018; Ribeiro et al., 2016; Shrikumar et al., 2017; Bach et al., 2015) involve locally approximating the machine learning model being explained with a simpler interpretable model. However, the usage of proxy models hinders the truthfulness of the explanations. Another explainability technique is visualizing the factors that influenced a model's decision through heatmaps (Fong et al., 2019; Elliott et al., 2021; Zhou et al., 2022). Heatmaps are useful to understand which objects present in the image have contributed to a classification. However, heatmaps do not show *how* areas of the image should be changed and they cannot explain factors that are not spatially localized (e.g., size, color, brightness, etc).

Explanation through examples or counterfactual explanations addresses these limitations by synthesizing alternative inputs (counterfactuals) where a small set of attributes is changed resulting in a different classification. These counterfactuals are usually created using generative models. A set of

Table 1: Comparison of explainers considered in this work. First column indicates whether counterfactuals are found with gradient descent. Second column displays the domain in which we perform the counterfactual search, with **z** referring to the attribute or latent space. Third column indicates whether the explainer takes into account changes in pixel space during optimization (e.g., visual similarity loss). Last column indicates if the explainer performs feature selection to generate counterfactuals.

| Method | Gradient based | Domain | Optimizes x-space | Feature selection |
|---|---|---|---|---|
| DiCE (Mothilal et al., 2020) | ✓ | z | ✗ | ✗ |
| DiVE (Rodríguez et al., 2021) | ✓ | z | ✓ | ✗ |
| GS (Laugel et al., 2017) | ✗ | z | ✗ | ✓ |
| StylEx (Lang et al., 2021) | ✗ | z | ✗ | ✓ |
| Latent-CF (Balasubramanian et al., 2020) | ✓ | z | ✗ | ✗ |
| xGEM (Joshi et al., 2018) | ✓ | z | ✓ | ✗ |

methods condition the generative model on attributes annotated in the dataset by using a conditional Generative Adversarial Network (GAN) (Joshi et al., 2018; Liu et al., 2019; Sauer & Geiger, 2021; Van Looveren et al., 2021; Yang et al., 2021). However, this approach restricts the explanations to the provided attributes which do not reflect the entirety of the image properties, making the applicability of these methods challenging where annotations are scarce. In order to generate counterfactuals without recurring to annotated attributes, another set of methods uses VAEs or unconditional GANs (Goodfellow et al., 2014a) that do not depend on attributes during generation (Rodríguez et al., 2021; Denton et al., 2019; Pawelczyk et al., 2020; Perez et al., 2018; Mothilal et al., 2020). See Table 1 for a comparison of the methods considered in our work.

**Explainability Benchmarks.** DiVE (Rodríguez et al., 2021) and DiCE (Mothilal et al., 2020) propose metrics that allow researchers to evaluate the quality of an explanation. These metrics evaluate the proximity of explanations to their original sample, and how diverse these are. Unfortunately, they are easy to game. For example, an explainer could maximize diversity by always modifying the same counterfactual attribute but randomly perturbing other non-counterfactual attributes to produce new redundant explanations. We propose a more general, harder to game metric that allows us to evaluate a set of explainers in order to identify their strengths and weaknesses through fair comparisons. Further, the set of attributes of a dataset can influence the evaluation of the explainability methods. CelebA (Liu et al., 2015a) is a common dataset used for generating counterfactual explanations (Rodríguez et al., 2021; Denton et al., 2019), and it is labeled with a series of attributes, such as "Atractive", that fail to fully describe the true underlying factors that generated the images (e.g, illumination, occlusions, contrast, etc). Likewise, there is no guarantee that unsupervised disentanglement methods such as VAEs identify the true factors of variations without making strong assumptions (Arjovsky et al., 2019). We sidestep these problems by evaluating all explainers in a common latent space with known attributes that fully describe the samples. Recently Pawelczyk et al. (2021) published a benchmark (CARLA) with an extensive comparison of several counterfactual explanation methods across 3 different tabular datasets. Our work differs from CARLA in three important ways: (1) we propose a principled metric to compare counterfactual explanation methods, (2) we introduce a new synthetic benchmark that allows comparing multiple explainers in a fair manner in the same latent space. (3) We focus on counterfactual visual explanations, which require access to a common latent space for fair comparison since pixel-level counterfactuals are difficult to interpret (e.g., adversarial attacks).

## 3 PROBLEM SETUP

In the following lines we describe a principled framework to quantify the quality of counterfactual explainers and show how it can be applied to compare multiple methods in the literature. In Section 3.1 we define the data generation process, in Section 3.2 we define the counterfactual generation process, in Section 3.3 we define the concept of optimal classifier used to compare the predictions of a model, and in Section 3.4 we define the metric used to evaluate counterfactual explanation methods. A notation table can be found in Table 3.

### 3.1 DATA GENERATION

Many explainability methods in the literature are designed for the image domain (Rodríguez et al., 2021; Joshi et al., 2018; Lang et al., 2021; Singla et al., 2019; Chang et al., 2018). In this area, most

datasets can be described with a data generating process where a set of latent variables ($\mathbf{z}$) result in an image ($x$) and a corresponding label (y), see Figure 1a. However, not all the latents that generate the image have an impact on the label ($\mathbf{z}_{\text{ind}}$). For example, the image brightness does not affect the presence of a dog. In addition, some latents can be correlated with the label ($\mathbf{z}_{\text{corr}}$). For instance, whenever there is a dog there is usually a dog collar. Formally, we consider a data generating process where a set of latent variables $\mathbf{z} \in \mathbb{R}^d$ are sampled from a prior $p(\mathbf{z})$, and a generator that produces images $p(x|\mathbf{z})$. Labels are generated using $p(y|\mathbf{z}_{\text{causal}})$, where $\mathbf{z}_{\text{causal}}$ is a subset of $\mathbf{z}$ containing direct causal parents of $y$ (Figure 1a). We also define $\mathbf{z}_{\text{corr}}$ as the set of attributes that are correlated to $y$ but not part of $\mathbf{z}_{\text{causal}}$.[1] Sometimes, these correlated attributes may have stronger predictive power, but relying on them would lead to unreliable predictions. For instance using the sky background for classifying airplanes. To generate datasets, we rely on a structural causal model (SCM) (Pearl, 2009), corresponding to a sequence of stochastic equations producing random variables based on the causal parents in the causal graph as described in Figure 1a.

In order to obtain a known mapping between $\mathbf{z}$ and $x$, we propose to leverage synbols (Lacoste et al., 2020), a synthetic dataset generator with many controllable attributes (font, character, color, rotation, size, *etc*). In addition, using a synthetic dataset allows us to control the effect of $\mathbf{z}$ on $x$ and specify the amount of change in $x$ relative to the amount of change in $\mathbf{z}$ (and vice-versa). Using synbols, we train an image generator $x = g(\mathbf{z})$[2], which is used to generate subsequent datasets. The generator $g$ is provided to the explainers to offer a differential mapping from $\mathbf{z}$ to $x$. We believe this is a strength of our benchmark compared to using datasets of natural images.

## 3.2 COUNTERFACTUAL GENERATION

Given an image $x$ and a classifier $\hat{f}(x)$, a counterfactual explanation method (explainer) produces $x'$, a perturbed version of $x$ that shows some insight about the sensitivity of $\hat{f}$ to the semantic attributes that describe $x$. The perturbation is commonly performed on a learned latent space $\mathbf{z}$. In general, explainers are tasked to learn an encoder and find a useful latent space, but this task is hard and still under active research. In order to bring a better comparison between explainers, we provide them access to the generating function $g$ and $\mathbf{z}$ so that explanations are generated in the same latent space. This gives us the opportunity to let explainers work directly in latent space by defining $\hat{h}(\mathbf{z}) := \hat{f}(g(\mathbf{z}))$. In other words, we define an explainer as:

$$\{\mathbf{z}'_i\}_{i=1}^n = e(\mathbf{z}, \hat{h}, g), \tag{1}$$

where $\mathbf{z}'_i$ is the $i$th counterfactual explanation from $\mathbf{z}$ found by explainer $e$ on the latent classifier $\hat{h}$. Working in latent spaces greatly simplifies the task of an explainer, but we will see that there are still a variety of challenges to be addressed. Namely, the notion of *optimal* classifier or *stable* classifier may be ill defined or may not always exists.

## 3.3 OPTIMAL CLASSIFIER

Counterfactual explanation methods tend to produce trivial explanations by perturbing the attribute being classified from the input (Rodríguez et al., 2021). A more useful explainer would change a model's predictions by perturbing non-causal attributes (such as the background of an image). To distinguish between these two kinds of explanations, an "oracle" is required, whose predictions are contrasted with those of the model. If an explanation changes both the oracle and the model's predictions, the explanation is deemed trivial and discarded. However, if the explanation only changes one of the two, the explanation is non-trivial. In the absence of a human oracle who knows the causal attribute being classified, the authors resort to an optimal predictor or *ground truth* classifier. However, the concept of optimal predictor is commonly ill defined and application dependant, thus we must proceed with care when assuming the existence of a ground truth classifier. To show that, we next define the concepts of Bayes classifier, causal classifier, and finally the causal classifier with non-reversible generator used in this work.

---

[1] $z \in \mathbf{z}_{\text{corr}}$ could be correlated to $y$ for two different reasons: i) $y \rightarrow z$ ii) a confounder $\alpha$ such that $y \leftarrow \alpha \rightarrow z$. Note that $\alpha$ may be element of $\mathbf{z}_{\text{causal}}$ or outside of the scene, such as the photograph.

[2] In this work, we consider deterministic generators. A more general formulation would be $g(x|\mathbf{z})$

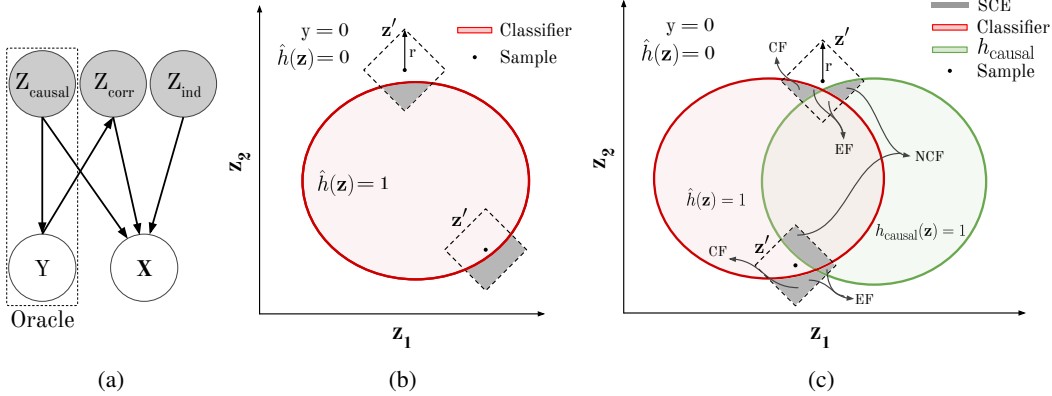

(a)          (b)          (c)

Figure 1: (a) Example of a causal graph satisfying the problem setup of section 3.1. (b) Successful counterfactual explanation as defined in (Mothilal et al., 2020; Joshi et al., 2018). That is, a successful counterfactual changes (gray) the classifier prediction (red) for the sample (point). The dashed square represents the maximum L1 norm of the perturbation performed by an explainer (c) Our definition of successful counterfactual explanation (gray) considers any change where an oracle (green) behaves differently from the classifier (red). EF: estimator flips, NCF: non-causal flips, CF: causal flips.

**Causal classifier with reversible generator.** The causal classifier makes predictions solely based on the causal parents of $y$ in the causal graph $G$. In latent space: $h_{\text{causal}}(\mathbf{z}) = \arg\max_y p(y|\mathbf{z}_{\text{causal}})$. When the generator $x = g(\mathbf{z}, \epsilon_x)$ is reversible, we obtain $f_{\text{causal}}(x) = h_{\text{causal}}\left(g^{-1}(x)\right)$. Interestingly, this classifier is robust to changes of $p(\mathbf{z})$ as long as $p(y|\mathbf{z}_{\text{causal}})$ and $p(x|\mathbf{z})$ remain unchanged.

**Causal classifier with non-reversible generator.** It is worth noting that when the generator is not reversible, a given $x$ can lead to many $\mathbf{z}$, which prevents from directly recovering $\mathbf{z}$ from $x$. A natural choice is to rely on the posterior distribution $f(x) = \sum_z p(\mathbf{z}|x)h_{\text{causal}}(\mathbf{z})$, where $p(\mathbf{z}|x) \propto p(\mathbf{z})p(x|\mathbf{z})$. However, this posterior now depends on $p(\mathbf{z})$, making the new classifier no longer independent to distribution shift when $p(\mathbf{z})$ is changed to e.g. $p'(\mathbf{z})$. This leads to the following negative result (see A.1 for the proof, along with an example):

**Proposition 1.** *If there exists a pair* $\mathbf{z}, \mathbf{z}'$ *s.t.* $g(\mathbf{z}) = g(\mathbf{z}')$ *and* $h_{\text{causal}}(\mathbf{z}) \neq h_{\text{causal}}(\mathbf{z}')$, *then for any deterministic classifier* $\hat{f}(x)$, *there is a prior* $p'(\mathbf{z})$ *s.t. the accuracy of* $\hat{f}$ *is 0 with respect to* $h_{\text{causal}}$.

### 3.4 EVALUATING COUNTERFACTUAL EXPLANATIONS

The goal for counterfactual generation methods is to find all the attributes that make a classifier behave differently from a causal classifier (see Figure 1c). Note that Mothilal et al. (2020) only considered counterfactuals that change the predictions of a classifier (Figure 1b), and Rodríguez et al. (2021) only considered the top region in Figure 1c. These definitions do not cover cases such as when the oracle changes its prediction while the classifier's stay the same. Following Mothilal et al. (2020); Rodríguez et al. (2021); Joshi et al. (2018), we also measure the similarity between the original example and the counterfactuals used to explain it. The reason is that counterfactuals should be relatable to original samples so that a human can interpret what is the sensitive semantic attribute. Next, we define the components of the proposed metric (Eq. 7).

**Proximal change (Joshi et al., 2018; Mothilal et al., 2020).** An explanation must be relatable to the original sample, thus it needs to be proximal. That is, the change $\mathbf{z}'$ needs to stay within a certain radius $r$ from $\mathbf{z}$. Using L1 norm, the set of proximal $\mathbf{z}'$ is defined as follows:

$$P_r(\mathbf{z}) = \{\mathbf{z}' \mid \|\mathbf{z} - \mathbf{z}'\|_1 \leq r\} \tag{2}$$

**Estimator Flip (EF) (Joshi et al., 2018; Mothilal et al., 2020).** This is defined as a proximal change on $\mathbf{z}$ leading to a change in prediction of the estimator $\hat{h}$ (see Figure 1b).

$$\text{EF}(\mathbf{z}) = \left\{\mathbf{z}' \;\middle|\; \hat{h}(\mathbf{z}') \neq \hat{h}(\mathbf{z})\right\} \cap P_r. \tag{3}$$

**Non-Causal Flip (NCF).** Counterfactuals obtained by estimator flips (EF) are common in the literature as they do not require the knowledge of $h_{\text{causal}}$. However, if we have access to $h_{\text{causal}}$, we can detect a new set of explanations: a proximal change in $\mathbf{z}'$ that changes the prediction of $\hat{h}$ but not of $h_{\text{causal}}$:

$$\text{NCF}(\mathbf{z}) = \{\mathbf{z}' \mid \text{EF}(\mathbf{z}) \ \wedge \ h_{\text{causal}}(\mathbf{z}') \ = \ h_{\text{causal}}(\mathbf{z})\} \cap P_r. \tag{4}$$

**Causal Flip (CF).** Additionally, access to $h_{\text{causal}}$ allows us to detect another new set of explanations: a proximal change in $\mathbf{z}'$ that changes the prediction of $h_{\text{causal}}$ but not $\hat{h}$:

$$\text{CF}(\mathbf{z}) = \left\{\mathbf{z}' \ \middle| \ \hat{h}(\mathbf{z}') \ = \ \hat{h}(\mathbf{z}) \ \wedge \ h_{\text{causal}}(\mathbf{z}') \ \neq \ h_{\text{causal}}(\mathbf{z})\right\} \cap P_r. \tag{5}$$

Thus, we define the set of successful counterfactual explanation (SCE) as follows:

$$\text{SCE}(\mathbf{z}) = (\text{NCF} \cup \text{CF}). \tag{6}$$

In summary, having knowledge of the causal factors (access to $h_{\text{causal}}$) allows us to evaluate counterfactuals explanations in a new way as illustrated in the following example. Given a dog classifier and an image of a dog, a counterfactual example that changes the background of the image in a way that alters the classifier's prediction (NCF) will almost certainly provide valuable insight about the model's behaviour. The same can be said about a counterfactual example that removes the dog from the image without altering the classifier's prediction (CF) (see Figure 1c). Note that these counterfactuals cannot be detected without causal knowledge, which is only available if we have access to the entire data generating process *i.e.,* a synthetic dataset.

**Orthogonal and complement subset.** Note that both EF and SCE are possibly infinite sets and cannot be easily interpreted by humans. We could return the explanation minimizing some notion of distance on $\mathbf{z}$ or $x$, however a good explainer should return a useful and diverse set of explanations.

To this end, we propose a metric that only takes into account the subset of orthogonal and complementary explanations. Otherwise, it is trivial to report many *explanations* that are a modification of an existing explanation without being useful. For instance, modifying the hair color to trigger a change in gender classification is a good finding, but changing the hair color again, and removing some clouds in the sky would not constitute a useful explanation. Hence, only admitting orthogonal explanations enforces a useful diversity. However, we also admit complementary explanations. That is, if darker hair triggers a change in gender classification and lighter hair also triggers a change, these are two useful explanations. In short, given two explainers that find counterfactuals by perturbing the most sensitive attribute, the orthogonality and complementary requirements ensure that the one that provides a more diverse set of counterfactuals by also perturbing less sensitive attributes scores higher. This is important because it rewards explainers that give a more complete description of the model to the user. There may be use cases where only the most sensitive attribute matters. However, everything else being equal, we argue that, in general, it is favorable to have access to a diversity of explanations.

The explainer is responsible for returning explanations produced with orthogonal or complementary perturbation vectors. To verify whether explanations are orthogonal or complementary we use a greedy algorithm. Concretely, we sort the explanations by how proximal they are to the original sample and add the first one to the set. Then we iterate through the rest and sequentially add every subsequent explanation that is orthogonal or complementary to all the explanations currently in the set (see Algorithm 1 for implementation). The resulting orthogonal and complement set is referred to as $SCE_\perp(\mathbf{z})$. We use the cardinality of this set to evaluate the performance of explainers:

$$\mathcal{S}_{\#} = |SCE_\perp(\mathbf{z})|. \tag{7}$$

We consider the proposed setup to be fairer than previous works, since: (1) all explainers are compared in the same latent space, resulting in a fair evaluation, (2) uninformative explanations are discarded leveraging knowledge of the causal factors, (3) it is designed to be more difficult to game by repeating counterfactual explanations, and (4) it rewards explainers that return a more complete set of explanations.

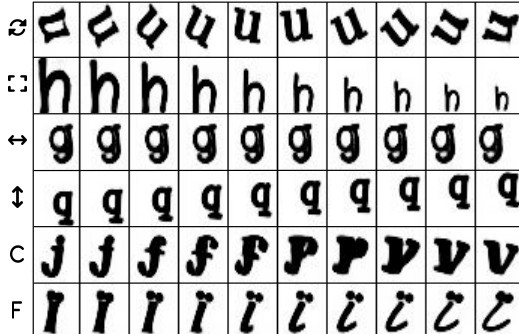

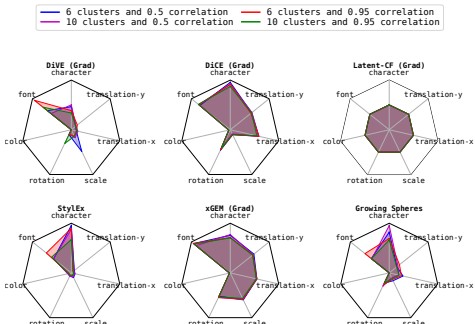

Figure 2: Interpolations produced by our learned generator ($g(\mathbf{z})$). Images are changed as the attribute's value changes smoothly from one value to another (left-right). From top to bottom: rotation, scale, h-translation, v-translation, char, font.

Figure 3: **Average attribute perturbation** for each method/scenario. Gradient based methods perturb almost all attributes while gradient-agnostic methods perturb only one or two. DiVE focuses almost solely on font.

## 4 EXPERIMENTS

In this section we give an overview of the different methods (see Table1) and datasets that are comprised within our benchmark. Since we provide access to a common interpretable latent space, we evaluate explainers that do not depend on a concrete latent decomposition. The code is written in PyTorch (Paszke et al., 2017) and will be made public. Implementations details can be found in the Appendix.

**Latent-CF** (Balasubramanian et al., 2020): A simple method that performs a adversarial perturbations in the latent space until a counterfactual with confidence higher than threshold $tol$ is found.

**DiCE** (Mothilal et al., 2020): A method that aims to produce a diverse set of counterfactual examples directly from a series of attributes or latent space by proposing a series of perturbations that change the predictions of a classifier. This is achieved by gradient-based optimization of multiple loss functions with respect to the attributes or latents and the classifier:

$$\mathcal{L} = \underbrace{\text{hinge\_loss}(\hat{h}(\mathbf{z}'), y, margin)}_{(A)} + \underbrace{\lambda_1 dist(\mathbf{z}, \mathbf{z}')}_{(B)} + \underbrace{\lambda_2 \text{dpp\_diversity}(\mathbf{z}')}_{(C)}, \quad (8)$$

where optimizing (A) pushes the prediction of the classifier $\hat{f}$ towards $y$ up to some margin, (B) ensures that counterfactuals ($\mathbf{z}'$) are close to the original samples ($\mathbf{z}$), and (C) maximizes the distance between each pair of counterfactuals.

**xGEM** (Joshi et al., 2018): A method equivalent to DiCE without the diversity term (C).

**DiVE** (Rodríguez et al., 2021): A method similar to DiCE that leverages the Fisher Information (FI) to find non-trivial counterfactuals, *i.e.* samples that change the classifier prediction without changing the causal attribute $\mathbf{z}_{\text{causal}}$, thus focusing on spurious correlations. This is done by masking out latent dimensions with the highest FI while optimizing a cost equivalent to Eq. 8.

**Growing Spheres (GS)** (Laugel et al., 2017): A method that given a data point $\mathbf{z}$ identifies its closest neighbour classified differently $\mathbf{e}$ referred to as *enemy*. This is done by finding the smallest $l_2$-ball around $\mathbf{z}$ that contains an *enemy*. Once $\mathbf{e}$ is found the dimensions with small changes in $\mathbf{e}$ with respect to $\mathbf{z}$ are discarded through a feature selection process, maximizing the sparsity of $\mathbf{e} - \mathbf{z}$.

**StylEx** (Lang et al., 2021)[3]: They find a latent perturbation in a direction that maximizes the difference in the output of the classifier for the original sample and its perturbed counterpart.

**Informed Search (IS)**: An explainer that knows about the data generation process in Figure 1a. Thus, IS generates explanations by perturbing the spuriously correlated attributes $\mathbf{z}_{\text{corr}}$.

---

[3]Since we already provide an interpretable set of latent attributes we evaluate only the Attribute finding (AttFind) algorithm from the paper

Table 2: Score (Eq. 7) and percentage of trivial counterfactuals () obtained by each explainer for each of the different datasets described in 4.1. Values represent an average score across batches for the entire dataset for 3 different runs.

| | Correlation | 0.50 | 0.95 | 0.50 | 0.95 |
|---|---|---|---|---|---|
| #Spurious | Explainer | $\mathcal{S}_{\#}$ | $\mathcal{S}_{\#}$ | Trivial (%) | Trivial (%) |
| 6 | IS (Oracle) 4 | 2.40 ±0.3 | 2.67 ±0.02 | 0.00 ±0.00 | 0.00 ±0.00 |
| | DiCE (Mothilal et al., 2020) | 1.18 ±0.01 | 1.17 ±0.01 | 9.37 ±0.11 | 6.78 ±0.26 |
| | DiVE (Rodríguez et al., 2021) | 1.02 ±0.00 | 1.00 ±0.01 | 2.51 ±0.09 | 1.68 ±0.02 |
| | GS (Laugel et al., 2017) | 1.01 ±0.00 | 1.01 ±0.00 | 4.49 ±0.40 | 2.34 ±0.15 |
| | StylEx (Lang et al., 2021) | 1.04 ±0.00 | 1.17 ±0.00 | 2.41 ±0.00 | 1.58 ±0.00 |
| | Latent-CF (Balasubramanian et al., 2020) | 0.82 ±0.00 | 0.93 ±0.00 | 0.00 ±0.00 | 0.00 ±0.00 |
| | xGEM (Joshi et al., 2018) | 1.18 ±0.02 | 1.15 ±0.01 | 12.46 ±0.03 | 6.45 ±0.07 |
| 10 | IS (Oracle) 4 | 2.80 ±0.04 | 3.63 ±0.02 | 0.00 ±0.00 | 0.00 ±0.00 |
| | DiCE (Mothilal et al., 2020) | 1.13 ±0.01 | 1.19 ±0.01 | 8.62 ±0.46 | 6.70 ±0.08 |
| | DiVE (Rodríguez et al., 2021) | 1.00 ±0.00 | 1.04 ±0.00 | 2.28 ±0.03 | 1.44 ±0.04 |
| | GS (Laugel et al., 2017) | 1.01 ±0.00 | 1.00 ±0.01 | 4.91 ±0.25 | 1.95 ±0.08 |
| | StylEx (Lang et al., 2021) | 1.15 ±0.00 | 1.12 ±0.00 | 3.37 ±0.00 | 1.62 ±0.00 |
| | Latent-CF (Balasubramanian et al., 2020) | 0.81 ±0.00 | 0.81 ±0.00 | 0.00 ±0.00 | 0.00 ±0.00 |
| | xGEM (Joshi et al., 2018) | 1.15 ±0.00 | 1.16 ±0.00 | 10.17 ±0.28 | 6.38 ±0.11 |

## 4.1 DATASETS

We design a synthetic benchmark based on the synbols dataset (Lacoste et al., 2020). In this benchmarks images are fully defined by 3 categorical attributes (48 fonts, 48 characters, 2 background colors) and 4 continuous attributes (x-translation, y-translation, rotation, scale), see Figure 2.

An advantage of synbols is the large amount of values in its categorical attributes such as character and font. This allows us to design different scenarios by introducing spurious correlations based on subsets of these attributes. From now on, we assume $\mathbf{z}_{\text{causal}} = \text{char} \in [1..48]$ and $\hat{h}_{\text{causal}} = \mathbf{z}_{\text{causal}}$ mod 2. Then we leverage the font attribute to introduce spurious correlations ($\mathbf{z}_{\text{corr}}$ in Figure 1a). Note that increasing the number of fonts in $\mathbf{z}_{\text{corr}}$ (the rest will be in $\mathbf{z}_{\text{ind}}$) increases the random chance of finding a counterfactual by accidentally switching the font. Likewise, increasing the amount of correlation between $\mathbf{z}_{\text{corr}}$ and $y$ makes spurious correlations easier to find since the classifier latches stronger on them. We hypothesize that stronger correlations will benefit gradient-based explainers, which will find higher gradient curvature for highly correlated fonts. To explore how explainers behave under different scenarios, we consider 6 and 10 spurious fonts with 50% and 95% correlation with $y$, resulting in a total of 4 scenarios. Further, we introduce a 5% of noise the the $\mathbf{z}_{\text{causal}}$ attribute (character) to encourage classifiers to also rely on the font.

## 4.2 RESULTS

We evaluate the performance of six different methods with the metric defined in Eq. 7. Each method is evaluated in several different datasets with varying levels of difficulty as described in 3.1 and 4.1. Additional results can be found in section A.4.

**It is hard to diversify.** A good explainer should be able to predict the behavior of the model with respect to changes in the different attributes that generate the data. In the case of a classifier, finding the attributes that induce it to change its prediction in order to reveal if it is relying on attributes that are independent from the class being predicted is a desirable goal. In the pursuit of this goal, an explainer should ideally populate SCE($\mathbf{z}$) (see Eq. 6) with explanations altering each of the attributes that are correlated with the data. However, as shown in Table 2 explainers fail to consistently find more than one altering attribute.

**Performance saturates with 6 fonts.** We observe that methods do not significantly increase the number of successful counterfactuals when adding 4 more spurious fonts (Table 2). This is, partially, because methods tend to focus on changing the $\mathbf{z}_{\text{causal}}$ attribute character as seen in Figure 3, which leads to trivial counterfactuals. When adding more fonts, the font identification task becomes more difficult for the classifier, which makes it more sensitive to characters and exacerbates this problem. For a more extensive ablation illustrating this phenomenon see Figure 5.

**Gradients tend to perturb most of the attributes.** Figure 3 offers insight into how each method perturbs $\mathbf{z}$ and we can see that gradient-based methods tend to perturb almost all attributes equally,

exploring the perturbation space in many directions. In the extreme, we found that Latent-CF slightly modifies all the latent attributes, producing counterfactuals that resemble adversarial attacks. While modifying all the attributes increases the chances of finding 1 good explanation on average, it also prevents the explainer from finding multiple non-trivial diverse explanations. On the other hand, methods that are gradient-agnostic focus on perturbing one or two attributes, resulting in a more narrow search space. This increases the risk of methods focusing on $\mathbf{z}_{\text{causal}}$ (Figure 1a). This is evidenced in Figure 3, where StylEx and GS considerably perturb the character attribute. Interestingly, the perturbation pattern of DiVE shares some similarities with StylEx and GS due to gradient masking.

**Explainers exploit bad classifiers.** As seen in Table 2 explainers are not significantly affected by the amount of spurious correlation $\mathbf{z}_{\text{corr}}$ introduced. This indicates that, in contrast with the oracle (IS), methods produce explanations by changing the font attribute $\mathbf{z}_{\text{causal}}$ (as seen in Figure 3) without changing the classifier's prediction, thus creating a successful counterfactual (Eq. 5). Figure 6 shows some examples of methods achieving this by changing the accent mark, the umlauts and the circumflex of vowel characters (e.g., ô → ö). These counterfactuals expose failure cases of the classifier and are therefore useful, since they show that the classifier is unable to classify some characters. See A.4 (**On causal counterfactuals**).

**DiVE focuses on changing the font.** As shown in Figure 3, DiVE perturbs almost *exclusively* the $\mathbf{z}_{\text{corr}}$ attribute (font), specially for high correlation values, this indicates that the method successfully distinguishes between $\mathbf{z}_{\text{causal}}$ and $\mathbf{z}_{\text{corr}}$ attributes. However, it is not able to consistently perturb the font in the right way to produce a diverse set of counterfactuals as evidenced by its score (Table 2).

**Non-triviality is not enough.** Table 2 (right) shows the average percentage of trivial counterfactuals found by each method. We observe that methods that tend to produce a higher number of successful explanations (left) tend to also produce a larger number of trivial counterfactuals (right), which are discarded in our metric.

## 5    DISCUSSION

**Benchmark** In this work, we have introduced a more comprehensive definition of good counterfactual (Section 3) that we instantiate as a metric (Section 3.4) as well as a fair evaluation setup (Section 3.1) in the form of a benchmark. Previous evaluation setups use datasets like CelebA where the causal data generation process is unknown and use metrics that are easy to game. In contrast, our evaluation setup uses a more comprehensive and fair metric while providing control over the entire data generating process and therefore knowledge of the causal factors. Even though knowledge of causal factors is rare when working in real world scenarios, it is possible to adapt our metric to take only into account an orthogonal and complement set of estimator flips EF (Eq. 3) which do not require causal knowledge. However, any evaluation schema that does not include causal information would be incomplete. Further, if an explainer fails to provide a set of useful and diverse explanations for our synthetic dataset it is very unlikely that it is able to do so for real datasets. Nevertheless, we recommend users to also evaluate explainers using real world data.

**Oracle** We argue that successful counterfactuals should be considered in the perspective of a human. In the absence of a human, we must resort to an optimal classifier, whose task is to contrast the predictions of the model with the optimal prediction and spot unexpected behaviors; acting as an oracle. Without an oracle, it is not clear how we could assess whether a model is working as intended. We show that the optimal classifier is commonly ill-defined in the image domain, because it is not always possible to access an invertible image generator (Section 3.3). Therefore, it cannot be expected that a classifier trained on pixel space achieves optimal performance.

**Results** Our experimental results indicate that the different counterfactual explainers in the literature perform similarly and there has been little improvement in the recent years (Table 2). Although most of them find a single explanation in average, we found that they do it in different ways (Figure 3).

## 6    CONCLUSION

In this paper we present a benchmark that provides unified metrics for evaluating different counterfactual explanation methods. The benchmark consists of synthetic images fully described by their annotated attributes which are accessible to the explainers through a differentiable generator. We hope this benchmark serves as an inspiration for developing future explainability methods.

## 7 REPRODUCIBILITY

In an effort to ensure reproducibility we provide implementation details in A.3. The total run-time for all experiments is ~37 hours on a single Titan-X GPU. The code[4] is written in PyTorch (Paszke et al., 2017) and is made public along with the datasets and pretrained weights for the models used in this work. Detailed documentation will be released upon publication.

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

# A  APPENDIX

## A.1  PROOF OF PROPOSITION 1

**Proposition 1.** *If there exists a pair* $\mathbf{z}, \mathbf{z}'$ *s.t.* $g(\mathbf{z}) = g(\mathbf{z}')$ *and* $h_{\mathrm{causal}}(\mathbf{z}) \neq h_{\mathrm{causal}}(\mathbf{z}')$, *then for any deterministic classifier* $\hat{f}(x)$, *there is a prior* $p'(\mathbf{z})$ *s.t. the accuracy of* $\hat{f}$ *is 0 with respect to* $h_{\mathrm{causal}}$.

*Proof.* Let $S = \left\{ \tilde{\mathbf{z}} \,\middle|\, \hat{f}(g(\tilde{\mathbf{z}})) \neq h_{\mathrm{causal}}(\tilde{\mathbf{z}}) \right\}$ and show that $|S| \neq 0$. Since $g(\mathbf{z}) = g(\mathbf{z}')$, we have that $\hat{f}(g(\mathbf{z})) = \hat{f}(g(\mathbf{z}'))$. Also, since $h_{\mathrm{causal}}(\mathbf{z}) \neq h_{\mathrm{causal}}(\mathbf{z}')$, we have either $\hat{f}(g(\mathbf{z})) \neq h_{\mathrm{causal}}(\mathbf{z})$ or $\hat{f}(g(\mathbf{z}')) \neq h_{\mathrm{causal}}(\mathbf{z}')$. Finally, any prior $p'(\mathbf{z})$ with no mass outside of $S$ satisfies the proof.  $\square$

As an example, let's consider a cube classifier from 2d projections of cubes. We can train a classifier to predict the correct class from the 3 visible faces. However, since the back of the cube is not visible there exists datasets such that the back of the cube is distorted in a way that makes it not a cube. The point being made here is that even though $h_{\text{causal}}$ exists, $f_{\text{causal}}$ does not always exists. Further assumptions needs to be made on which priors $p(\mathbf{z})$ are valid for robustness to distribution shift, e.g., as humans, we make a symmetry assumption for classifying cubes. Perhaps future line of research in explainability should seek to find what assumptions are made by classifiers. In the rest of this work, we will make the assumption that if $g(\mathbf{z}) = g(\mathbf{z}')$, then $h_{\text{causal}}(\mathbf{z}) = h_{\text{causal}}(\mathbf{z}')$ for all pairs $\mathbf{z}, \mathbf{z}'$ [5]

The proof is simple but it contrasts with the known result that $h_{\text{causal}}$ is robust to any distribution shift over $p(\mathbf{z})$.

## A.2    Information of the Explanations

In our work, we aim to improve the amount of information provided by explanation sets. In terms of information theory, unexpected events tend to be more informative. In this sense, repeating the same explanation multiple times provides little information and increasing diversity increases the amount of information. In addition, the amount of information of an event is usually measured with respect to some underlying probability distribution. If we know that a classifier is trained to detect a certain object, we expect the classifier to learn the conditional probability distribution (label given input) of the training set. Thus, deviations from this distribution are unexpected and informative. However, we do not know the real label for new samples and thus, we need to resort to some form of oracle or ground truth classifier in order to compare it with the model being explained. This ground truth classifier must have access to the real data generating process in order to infer the correct class from images, and the data generating process of image-label distributions is typically a causal process governed by the laws of physics. That is why we refer to this classifier as "causal classifier".

## A.3    Implementation Details

In this section we will describe the training setting for the encoder $q(\mathbf{z}|x)$ the generator $g(\mathbf{z})$ and the classifier to be explained $\hat{f}(x)$, along with the settings for each of the explainers considered in this work.

### A.3.1    Training details

**Encoder** The encoder is based on BigGAN's (Brock et al., 2018; Rodríguez et al., 2021) discriminator architecture with a classifier on top and it is trained on Synbols (Lacoste et al., 2020) dataset. Given an image $x$ we task the encoder with predicting the attributes that describe it. It is trained for 100 epochs with a batch size of 64. We use AdamW (Loshchilov & Hutter, 2017) with a learning rate of 0.001 and a weight decay of 0.0001 with a cosine annealing  learning rate scheduler(Loshchilov & Hutter, 2016). Since Synbols contains discrete and continuous attributes, we optimize them separately minimizing:

$$\mathcal{L}_{\text{discrete}} + \mathcal{L}_{\text{continuous}} \tag{9}$$

where $\mathcal{L}_{\text{discrete}}$ is the average cross entropy loss for each categorical attribute and $\mathcal{L}_{\text{continuous}}$ is the $L_1$ distance between the original continuous attributes and the ones predicted.

**Generator** Similar to the encoder, the generator is also based on a BigGAN (Brock et al., 2018) architecture and trained with the same hyper-parameters and learning rate scheduler. Given a set of attributes $\mathbf{z}$ and the image $x$ that they describe, we train the generator to reconstruct $x$ from $\mathbf{z}$. However, in order to provide a high-dimensional input to the generator instead of directly using the 7 Synbols attributes leveraged in our benchmark $z \in \mathbb{R}^7$ (Section 4.1), we use embedding layers to project each categorical attribute (character and font) into a 3-dimensional space. The embeddings are concatenated with the 5 continuous[6] Synbols attributes obtaining $\mathbf{z} \in \mathbb{R}^{11}$. We train the generator minimizing the following criteria:

---

[5]This assumption could be relaxed by saying that the probability of this assertion being violated is unlikely under a predefined set of *valid* distribution shifts.

[6]We treat the background color as a continuous attribute

Table 3: Notation Table.

| Notation | Description |
|---|---|
| $x$ | An input image |
| $y$ | The label of the input image |
| $\mathbf{z}$ | A set of latent variables |
| $\mathbf{z}_{\text{causal}}$ | A subset of $\mathbf{z}$ containing the causal parents |
| $\mathbf{z}'$ | A counterfactual explanation for $\mathbf{z}$ |
| $\hat{f}(x)$ | The classifier to be explained |
| $\hat{h}(\mathbf{z})$ | The latent classifier to be explained |
| $h_{\text{causal}}(\mathbf{z})$ | The latent classifier to that is robust under change of $p(\mathbf{z})$ |
| $p(\mathbf{z})$ | Prior distribution over $\mathbf{z}$ |
| $g(\mathbf{z})$ | Deterministic image generator |
| $g(x|\mathbf{z})$ | Stochastic image generator |
| $q(\mathbf{z}|x)$ | An encoder |
| $e(\mathbf{z}, \hat{h}, g)$ | A function that generates explanations |

$$\mathcal{L}_{rec} = \alpha \times \| x - x' \|_1 + (1 - \alpha) \times \| q(x) - q(x') \|_1 \qquad (10)$$

where $\alpha = 0.2$, $x$ and $x'$ are the original and reconstructed image respectively and $q(x)$ are the *learned* encoder features for image $x$. Lastly, to reduce the amount of noise in the generated images we use a discriminator network $D$ trained alongside the generator to distinguish between $x$ and $x'$. Specifically the generator is trained every 3rd iteration. So, the criteria for the generator to optimize becomes:

$$\mathcal{L}_{\text{rec}} + \log(1 - D(x')) * \lambda \qquad (11)$$

where $\lambda = 0.01$ and $D(x')$ is the discriminator's estimate of the probability that the reconstructed image $x'$ is real.

**Classifier** The classifiers we set to explain are ResNet-18 (He et al., 2016) architectures trained on the different benchmarks described in Section 4.1. All the classifiers are trained for 10 epochs with a batch size of 256. We use AdamW with a learning rate of 0.01 and a weight decay of 0.0001 with a cosine annealing learning rate scheduler.

### A.3.2 EXPLAINER EVALUATION

In this section we will detail the hyper-parameters chosen for each of the methods analyzed in this work as well as some details regarding the benchmark. Given the varied nature of the explainers and their hyper-parameters we will refer to the works where the explainers are introduced when describing the effect the hyper-parameters have on each method. All of the hyper-parameters where chosen through random search.

**Common setup** Across our experiments there are a few settings that are common for all methods. To save time, instead of producing counterfactuals for the entire validation dataset we select a balanced subset with a total of 800 correctly and incorrectly classified samples with different levels of confidence. We do this by selecting the 100 samples closest to the required level of confidence $\hat{f}(x) \in (\pm 0.1, \pm 0.4, \pm 0.6, \pm 0.9)$.

The explainers are required to produce 10 counterfactuals per sample, if a method is originally conceived to produce only one, we follow (Mothilal et al., 2020) to create 10 samples close to the original sample in $\mathbf{z}$ space and task the method with producing an explanation for each of them. The attributes in $\mathbf{z}$ are standardized using the mean and standard deviation of the attributes in the training set. To prevent explainers from generating counterfactuals that the generator $g(\mathbf{z})$ cannot interpret we clip every coordinate in $\mathbf{z}$ to the maximum and minimum values for each attribute found in the training set. We set the batch size to 12 across experiments.

---

**Algorithm 1** Generating an orthogonal set of counterfactuals

```
def orthogonal_set(z:np.ndarray, e_sc: np.ndarray, tau=0.15: float) -> np.ndarray: # d X n,d X 0 -> n,d
    """
    Receives the latents of an original sample (z), and a set of successful counterfactuals
    (e_sc from Eq. 6) and returns an orthogonal set of counterfactuals (Eq. 7).
    Two samples z1 and z2 are orthogonal whenever abs(cos(z1, z2)) < tau.
    Two samples z1 and z2 are complementary whenever cos(z1, z2) + 1 < tau
    """
    z = z[None, :] # d -> 1,d
    delta_sc = e_sc - z # compute perturbation vector
    delta_sc_norm = np.linalg.norm(delta_sc, 1, axis=1) # compute norm of perturbations
    indices = delta_sc_norm.argsort()
    delta_orth = delta_sc[None, indices[0]] # initialize set of orthogonal perturbations
    for i in indices[1:]:
        sim = cos(delta_sc[i], delta_orth) # calculate cosine similarity wrt all ellements in a set
        cond_orth = (np.abs(sim) < tau).all() # orthogonality condition
        cond_comp = (sim + 1 < tau).any() # complementary condition
        if cond_orth or cond_comp: # if condition satisfied, add to set
            delta_orth = np.concatenate([delta_orth, delta_sc[None, i]], axis=0)
    return z + delta_orth # return set of orthogonal counterfactuals
```

---

Since the categorical attributes in $\mathbf{z}$ space (font and character) are produced by embedding layers and projected into a 3-dimensional space each (Section A.3.1), they must be treated differently from the continuous attributes. When measuring if an explanation $\mathbf{z}'$ is proximal to the original sample $\mathbf{z}$ we establish three different radius $r$ values (see Eq. 2), one for the font, one for the character and one for the continuous attributes. For the character and font attributes we set $r$ to be the maximum pairwise distance between the embedding representations of each attribute and for continuous attributes we set $r = 1$. Making this distinction allows the explainer to change from any given font/character to another while preventing it from modifying every attribute at once. When verifying if counterfactuals fulfill our orthogonality condition we look at perturbations between continuous attributes ($\mathbf{z}'_{\text{cont}}$). However, for embedded attributes (font and character) we transform their embedded representation in $\mathbf{z}$ space into:

$$\mathbf{z}'_{\text{cat}} = \text{Softmax}(\| \mathbf{c}' - \mathbf{w}_i \|_2 \,/\, -t), \forall i \in [1..48] \tag{12}$$

where $\mathbf{c}'$ is the representation of the categorical attribute character/font for a given counterfactual $\mathbf{z}'$, $\mathbf{w}$ are the weights of the embedding layer used to map the 48 characters/fonts to their 3 dimensional representation $\mathbf{c}'$ and $t$ is a temperature factor set to $0.33$. We choose the softmax function so that any small change in $\mathbf{c}'$ that maps to a different categorical attribute can be orthogonal. In order to turn $\mathbf{z}'_{\text{cat}}$ into a perturbation vector we cancel the perturbation on the original attributes coordinate by setting:

$$\mathbf{z}'_{\text{cat}} = \mathbf{z}'_{\text{cat}} \times (\mathbf{1} - \mathbf{1}_c) \tag{13}$$

where $\mathbf{1}$ is a vector of 1s and $\mathbf{1}_c$ is a one-hot representation of the categorical attributes for the original sample. Thus given a sample $\mathbf{z}$ we verify that two counterfactual explanations $\mathbf{z}'$ and $\mathbf{z}''$ are orthogonal by measuring cosine similarity: $cos(\mathbf{z}'_{\text{cat}}, \mathbf{z}''_{\text{cat}})$ for categorical attributes and $cos(\mathbf{z}'_{\text{cont}} - \mathbf{z}, \mathbf{z}''_{\text{cont}} - \mathbf{z})$ for continuous attributes.

**Growing Spheres (GS) (Laugel et al., 2017)** For this method we found the best configuration was setting the initial radius $\eta$ to 10 and the number of candidates $n$ to 50. For a detailed description of the role of these hyper-parameters see (Laugel et al., 2017).

**StylEx (Lang et al., 2021)** For this method we found the best configuration was setting threshold $t$ to $0.3$ using the "Independent" selection strategy and the amount of shift applied to each coordinate to $0.8$. This last parameter is not mentioned in the Stylex (Lang et al., 2021) paper, but it can found as a parameter under the name *shift_size* in the implementation they provide. For a detailed description of the role of these hyper-parameters see (Lang et al., 2021).

**DiCE (Mothilal et al., 2020)** For this method the best results were obtained by setting to learning rate to $0.1$, the reconstruction weight of the loss function $\lambda_1$ to 1 and the diversity weight of the loss function $\lambda_2$ to 1. For a detailed description of the role of these hyper-parameters see (Mothilal et al., 2020). We set the maximum number of iterations to 50 in case the algorithm does not converge.

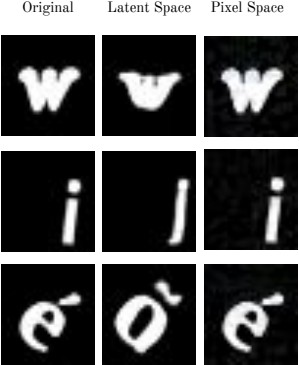

Figure 4: Comparison of perturbations performed in latent and pixel space. Note how the perturbations performed in latent space convey meaningful information (e.g., changes in the font or character), while the ones performed in pixel space resemble adversarial attacks and do not provide any valuable insights into the classifier's reasoning.

**DiVE (Rodríguez et al., 2021)** For this method we found the best configuration was setting the learning rate to 0.1, the weight of the proximity loss term to 0.0001, the factor that controls the sparsity of the latent space $\gamma$ to 0.1 and the weight of the diversity loss to 0.001. For a detailed description of the role of these hyper-parameters see (Rodríguez et al., 2021). We set the maximum number of iterations to 50.

**xGEM (Joshi et al., 2018)** This method shares some parameters with DiVE. We found the best configuration for this explainer was setting the learning rate to 0.1 and the weight of the proximity loss term to 0.001. We set the maximum number of iterations to 50.

**Latent-CF (Balasubramanian et al., 2020)** For this method we set the learning rate to 0.1, the probability of target counterfactual class $p$ to 0.1 and the tolerance $tol$ to 0.5. For a detailed description of the role of these hyper-parameters see (Balasubramanian et al., 2020). We set the maximum number of iterations to 50 in case the algorithm does not converge.

### A.4 ADDITIONAL RESULTS

In this section we present some qualitative results in the form of counterfactuals generated by each method, along with extra quantitative results shown in Table 4. We also provide an example illustrating why explanations found in pixel space are uninterpretable (Figure 4).

**Quality over quantity** As seen in Table 4 some explainers obtain a high percentage of successful counterfactuals SCE, sometimes even higher than the oracle (xGEM, DiVE). However, this is not

Table 4: From left to right: We report the percentage of estimator flips EF (Eq. 3) (Joshi et al., 2018; Mothilal et al., 2020), percentage of successful counterfactuals SCE (Eq. 6) and what percentage of those are Causal Flips and Non-Causal Flips (see Section 3.4 and Eq. 6). Values represent an average score across batches for the entire dataset for 3 different runs.

| | Correlation | 0.50 | 0.95 | 0.50 | 0.95 | 0.50 | 0.95 | 0.50 | 0.95 |
|---|---|---|---|---|---|---|---|---|---|
| #Spurious | Explainer | EF (%) | | SCE (%) | | Causal Flip Rate (%) | | Non-Causal Flip Rate (%) | |
| 6 | IS (Oracle) | $40.55_{\pm0.16}$ | $76.97_{\pm0.24}$ | $67.5_{\pm4.40}$ | $62.25_{\pm4.45}$ | $0.00_{\pm0.00}$ | $0.00_{\pm0.00}$ | $100_{\pm0.00}$ | $100_{\pm0.00}$ |
| | DiCE (Mothilal et al., 2020) | $32.90_{\pm0.05}$ | $31.96_{\pm0.11}$ | $55.42_{\pm2.60}$ | $56.67_{\pm2.88}$ | $26.46_{\pm1.56}$ | $29.22_{\pm1.20}$ | $73.54_{\pm1.56}$ | $70.78_{\pm1.20}$ |
| | DiVE (Rodríguez et al., 2021) | $25.02_{\pm0.38}$ | $36.58_{\pm0.12}$ | $67.5_{\pm1.25}$ | $60.83_{\pm2.60}$ | $6.55_{\pm0.28}$ | $3.44_{\pm0.15}$ | $93.45_{\pm0.28}$ | $96.56_{\pm0.15}$ |
| | GS (Laugel et al., 2017) | $36.14_{\pm1.22}$ | $34.94_{\pm0.49}$ | $31.67_{\pm7.10}$ | $31.67_{\pm15.63}$ | $0.00_{\pm0.00}$ | $0.00_{\pm0.00}$ | $100_{\pm0.00}$ | $100_{\pm0.00}$ |
| | Stylex (Lang et al., 2021) | $23.66_{\pm0.00}$ | $24.84_{\pm0.00}$ | $23.07_{\pm0.00}$ | $25.80_{\pm0.00}$ | $17.02_{\pm0.00}$ | $21.40_{\pm0.00}$ | $82.98_{\pm0.00}$ | $78.60_{\pm0.00}$ |
| | Latent-CF (Balasubramanian et al., 2020) | $20.98_{\pm0.00}$ | $24.18_{\pm0.00}$ | $20.96_{\pm0.00}$ | $24.18_{\pm0.00}$ | $0.00_{\pm0.00}$ | $0.00_{\pm0.00}$ | $100_{\pm0.00}$ | $100_{\pm0.00}$ |
| | xGEM (Joshi et al., 2018) | $70.61_{\pm0.28}$ | $75.98_{\pm0.25}$ | $76.67_{\pm9.21}$ | $78.33_{\pm1.90}$ | $4.18_{\pm0.69}$ | $2.96_{\pm0.22}$ | $95.82_{\pm0.00}$ | $97.04_{\pm0.00}$ |
| 10 | IS (Oracle) | $35.33_{\pm0.16}$ | $71.19_{\pm0.08}$ | $54.50_{\pm2.43}$ | $51.25_{\pm1.25}$ | $0.00_{\pm0.00}$ | $0.00_{\pm0.00}$ | $100_{\pm0.00}$ | $100_{\pm0.00}$ |
| | DiCE (Mothilal et al., 2020) | $31.77_{\pm0.45}$ | $33.25_{\pm0.23}$ | $45.00_{\pm4.33}$ | $39.17_{\pm0.72}$ | $26.16_{\pm3.43}$ | $27.65_{\pm0.79}$ | $73.84_{\pm3.43}$ | $72.35_{\pm0.79}$ |
| | DiVE (Rodríguez et al., 2021) | $22.39_{\pm0.31}$ | $31.8_{\pm0.25}$ | $60.00_{\pm1.25}$ | $54.58_{\pm0.72}$ | $6.83_{\pm0.57}$ | $2.25_{\pm0.01}$ | $93.17_{\pm0.57}$ | $97.75_{\pm0.01}$ |
| | GS (Laugel et al., 2017) | $37.38_{\pm0.38}$ | $36.38_{\pm0.54}$ | $44.58_{\pm12.52}$ | $40.42_{\pm5.90}$ | $0.00_{\pm0.00}$ | $0.00_{\pm0.00}$ | $100_{\pm0.00}$ | $100_{\pm0.00}$ |
| | Stylex (Lang et al., 2021) | $24.20_{\pm0.00}$ | $23.04_{\pm0.00}$ | $23.65_{\pm0.00}$ | $23.77_{\pm0.00}$ | $21.60_{\pm0.00}$ | $20.35_{\pm0.00}$ | $78.40_{\pm0.00}$ | $79.65_{\pm0.00}$ |
| | Latent-CF (Balasubramanian et al., 2020) | $22.00_{\pm0.00}$ | $23.33_{\pm0.00}$ | $21.98_{\pm0.00}$ | $23.24_{\pm0.00}$ | $0.00_{\pm0.00}$ | $0.00_{\pm0.00}$ | $100_{\pm0.00}$ | $100_{\pm0.00}$ |
| | xGEM (Joshi et al., 2018) | $66.45_{\pm0.63}$ | $74.95_{\pm0.30}$ | $61.67_{\pm5.20}$ | $62.92_{\pm5.90}$ | $4.51_{\pm0.35}$ | $2.15_{\pm0.22}$ | $95.49_{\pm0.00}$ | $97.85_{\pm0.27}$ |

reflected in their score $\mathcal{S}_{\#}$ (Table 2), which is considerably lower than the oracle's. This is because even though the explainers can find a high number of counterfactuals they are discarded by our metric since they are not orthogonal or complementary and thus redundant. Further, note that the score measured using estimator flips EF (Eq. 3) (Joshi et al., 2018; Mothilal et al., 2020) is not correlated with our score $\mathcal{S}_{\#}$ (Table 2). For example, xGEM obtains a higher score than the oracle (IS) despite the latter returning a more complete set of explanations. This shows how previously proposed metrics (Joshi et al., 2018; Mothilal et al., 2020) can be gamed by explainers by generating many redundant explanations that fail to fully the describe the model's behaviour. Figure 5, also supports this finding, showing how the performance of the oracle (IS) is the only one affected by the amount of correlated attributes and their level of correlation (see Section 4.1).

**On causal counterfactuals** As seen in Table 4, some methods produce a high amount of causal change counterfactuals *i.e.,* they change $\mathbf{z}_{\text{causal}}$ and create a counterfactual that does not change the classifier's prediction whilst changing the oracle's prediction, exposing failure cases in the classifier. We find that DiCE (Mothilal et al., 2020) and StylEx (Lang et al., 2021) produce a high amount of these counterfactuals, while GS (Laugel et al., 2017) and Latent-CF (Balasubramanian et al., 2020) always change the classifiers prediction and thus produce none. The oracle (IS) is not designed to perturb $\mathbf{z}_{\text{causal}}$ in any way so it cannot produce any causal counterfactuals. Figure 6 shows two ways in which explainers achieve these counterfactuals. We can see that explainers that generate a high amount of causal counterfactuals (DiCE, StylEx) can be very "creative", when modifying the character. Note that besides confusing the classifier by modifying the character's diacritic, explainers can create new characters entirely by merging two letters together or even adding an accent mark to a consonant. Note that this behaviour is unavoidable in the absence of an optimal classifier.

**Additional insights** As seen in Table 2 explainers are unable to generate a diverse set of counterfactual explanations. However, Table 4 highlights some differences between methods when it comes to other metrics. If the objective is to maximize the number of estimator flips EF or the number of successful counterfactuals SCE we recommend using xGEM. If the objective is to maximize the number of causal flips we recommend using StylEx or DiCE. Note that explainers generate a high amount of redundant counterfactuals, and choosing them based on how they maximize these individual metrics is of little use.

## A.5 LIMITATIONS

As discussed in Section 3.3, the core limitation of explaining image classifiers via latent perturbations is the lack of accurate reversible generators. If the generator is not reversible, a given $x$ can lead to many $\mathbf{z}$, which prevents the direct recovery of $\mathbf{z}$ from $x$. It might be a good idea to bypass the

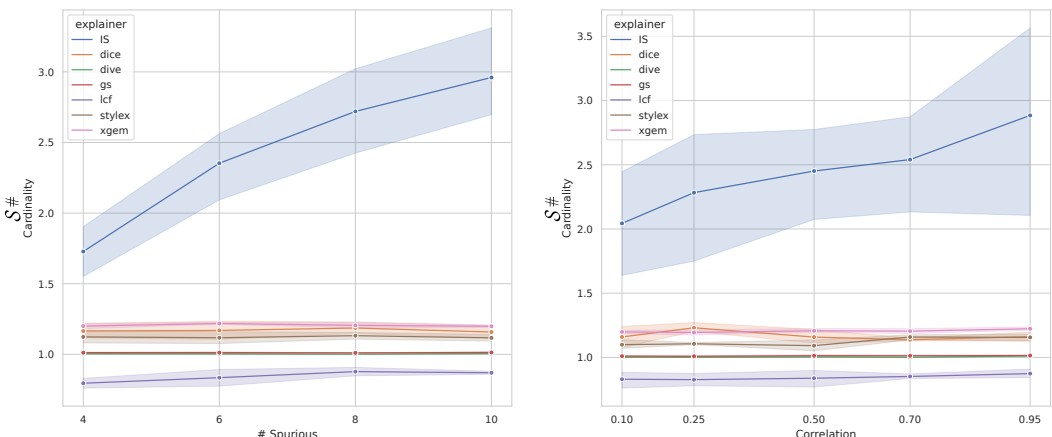

Figure 5: Sensitivity of every explainer to varying amount of correlation levels (left) and number of spuriously correlated attributes (right) (see Section 4.1) measured with our score (Eq. 7). Note how the performance of the explainers, excluding the oracle (IS), does not scale and is not sensitive to the level of correlation and the amount of correlated variables. This supports our findings that explainers are unable to provide a diverse set of explanations and focus on changing the causal attribute $\mathbf{z}_{\text{causal}}$.

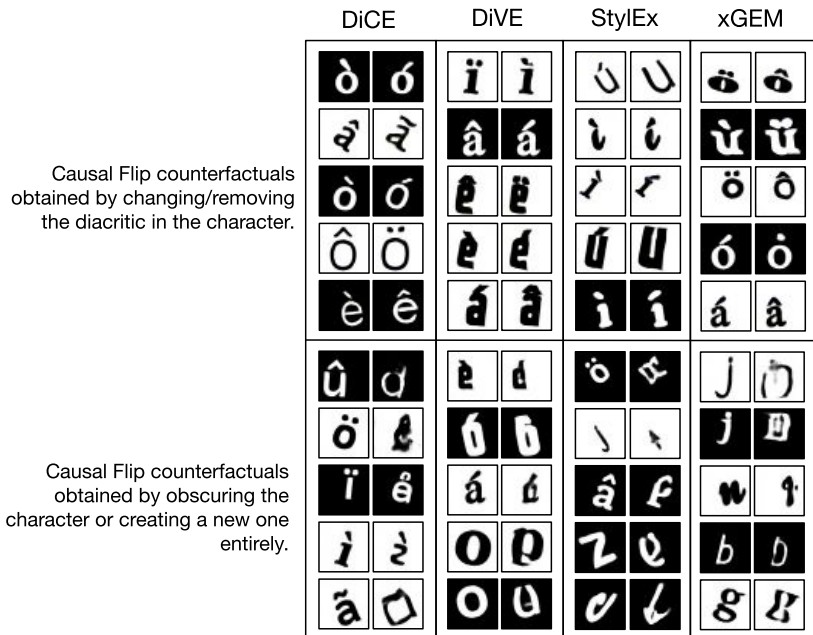

Figure 6: Some of the Causal Flip counterfactuals (see Section 3.4 and Eq. 6) obtained by each method separated into two different subcategories.

pixel space completely and work directly on **z**, this however, would produce explanations outside the image domain and therefore, uninterpretable by humans, which is ultimately not very useful. It could be argued that the generator used in this work could be modified to yield better image reconstructions given any **z**, however this will always be hindered by the aforementioned limitation. More generally, most methods rely on some some sort of latent decomposition in order to search for counterfactuals in a latent space. However, it is still not clear how the true latent variables of the data generating process are not identifiable (Locatello et al., 2019). In this work we circumvent this problem by using a synthetic dataset. On the other hand, Khemakhem et al. (2020) showed that, with further assumptions, it is possible to identify the latent variables (Khemakhem et al., 2020). Moreover, in a temporal setup, it is possible to identify which of these latent variables are the causal ones (Lachapelle et al., 2022). Finally, in a multi-task setup where distribution shift occurs, it is possible to identify which variables are robust to distributions shift and hence, likely to be the causal ones. In summary, although it would be possible to approximate the true latent factors in some cases, it would require making additional assumptions about the data.

We make an effort to establish a fair, principled metric that is useful. However, this metric does not depict all the properties of an explainer such as fragility or speed. It is possible, albeit unlikely, that our definition of useful/informative explanation might not always align with that of the user. Lastly, the generator we use in this work can generate images with certain implicit biases. However, please note that our benchmark is a tool to evaluate properties of explainers that are impossible to evaluate without a synthetic setup.

## A.6 ETHICAL CONCERNS

Research in explainable AI is crucial for safe deployment of machine learning solutions in real life. In this work, we have shown that the lack of a principled evaluation of such systems has slowed-down advancements in the field. These findings should not discourage future research in explainable AI, on the contrary, we hope that our work unblocks the current state of the art and spurs further progress.

On the other hand, the research described in this paper does not (i) directly facilitate injury to living beings, (ii) raise safety, privacy or security concerns, (iii) raise human rights concerns, (iv) have a

detriment effect on people's livelihood or economic security, (v) develop or extend harmful forms of surveillance, (vi) severely damage the environment, or (vii) deceive people in ways that cause harm.

Lastly, our research does not use human-derived data, or has involved extensive annotation by human research participants. The synthetic dataset used in this paper has not been deprecated for technical, legal, or ethical reasons.

