# OpenReview forum: "Evaluating Counterfactual Explainers"
_ICLR.cc/2023/Conference — Submitted to ICLR 2023_

### Official Review · Reviewer_pgaK · 2022-10-20

**Confidence:** 4
**Correctness:** 2
**Technical Novelty And Significance:** 2
**Empirical Novelty And Significance:** 2
**Recommendation:** 3

**Clarity, Quality, Novelty And Reproducibility:**

I think the paper is based on some interesting and novel ideas. However, the contextualization, motivation, explanation, evaluation, and presentation of the proposed measures are rather weak and raise many questions and remarks:

1. Are the proposed measures discriminative and meaningful for evaluation (given that there is little variance in the results)?
2. Do the results generalize to other datasets or other similar generative processes?
3. In general, different explanation techniques use the input space directly or the latent space to identify counterfactual explanations. The authors state that they use the latent space representation for all evaluated methods, which would also mean that the methods under consideration then rely on a similar optimization.  If this is true, then it is quite unusual as some of the counterfactual algorithms first learn a generative model, and then find counterfactuals with respect to the learnt model. This could also be one of the reasons why there is so little variation across the diversity results.
4. The data generating process from Figure 1(a): Why is there an arrow from Y to Z_corr? Is it intended that X \independent Y | Z?
5. The title suggests that this is a general benchmark for counterfactual explanation algorithms; however, the authors only focus on counterfactual explanations for image data, and it would make sense to reflect this in the title.
6. How does Proposition 1 fit into your benchmarking narrative? Could the authors explain that more clearly?
7. In section 3.4 you say that “the goal of counterfactual generation methods is to find all the attributes that make a classifier behave differently from a causal classifier”. Since in practice we never usually know the causal data generating process, I am wondering whether this is really the goal of counterfactual generation methods. How would one evaluate this the achievement of this goal in practice?

As it can be seen, there are many points that should be clarified or better described and motivated. Also, some real-world examples could help improve the presentation.


**Strength And Weaknesses:**

Strengths:
1. The addressed questions are timely and interesting to the XAI community.
2. The proposed idea for deriving evaluation measures for counterfactual explanations is sensible.

Weaknesses:
1. The generative model and specifically the dependencies represented in the model (Fig. 1 (a)) are not well-motivated.
2. The evaluation is weak: (a) the authors use only one (black-and white) synthetic dataset, (b) the different evaluation choices are not well motivated, (c) it is not clear which results generalize to other settings, (d) the evaluated CE methods behave similarly according to the proposed measure (Eq. 7), so either the authors evaluate very similar methods, or the proposed measures are non-discriminative.
3. The structure and the clarity of the paper could be considerably improved (see points below).

**Summary Of The Paper:**

The authors suggest a data generating process to evaluate the quality of counterfactual explanations with respect to factors of variation and propose two evaluation measures. The goal of the evaluation measures is to evaluate to what extent different explanation methods are (1) successfully changing the predicted label to the target label, and (2) to measure to what extent these methods generate diverse counterfactual explanations.
For their first measure the authors propose to use a generative model, where they model, three latent factors (causal, correlational and independent factors). While all three latent factors influence the observed features (X), only the causal latent factors truly influence the target variable (Y). Using this data generating process the authors define a successful counterfactual explanation to be one which relies on the causal as opposed to the correlational or independent features.
The second evaluation measure counts how many orthogonal or complementary explanations were output by any of the counterfactual explanation methods. The authors suggest that diversity of counterfactual explanations is best accessed by non-trivially diversifying explanations. To achieve this, the authors suggest using orthogonality between any two explanation vectors as a measure for non-trivial diversity.
Finally, the authors use their diversity measure to evaluate the performance of 6 different counterfactual explanation methods on a synthetic dataset.


**Summary Of The Review:**

Overall, the paper presents some interesting and novel ideas, which unfortunately are not well presented and evaluated.

---

> ### Author Response · Authors · 2022-11-12
> **Response to reviewer pgaK (2/2)**
>
> **In general, different explanation techniques use the input space directly or the latent space to identify counterfactual explanations. The authors state that they use the latent space representation for all evaluated methods, which would also mean that the methods under consideration then rely on a similar optimization. If this is true, then it is quite unusual as some of the counterfactual algorithms first learn a generative model, and then find counterfactuals with respect to the learnt model. This could also be one of the reasons why there is so little variation across the diversity results.**
>
> We only considered methods that do not rely on a concrete latent decomposition. In the case of Stylex which learns a GAN to identify the latent factors, we evaluate only the Attribute finding (AttFind) algorithm from the paper, since we already provide an interpretable set of latent attributes. This is an important point and we have clarified it in Section 4.
>
> **The data generating process from Figure 1(a): Why is there an arrow from Y to Z_corr?**
>
> Because there are attributes that influence the presence of other attributes. For instance, wherever there is a dog there is usually a dog collar.
>
> **The title suggests that this is a general benchmark for counterfactual explanation algorithms; however, the authors only focus on counterfactual explanations for image data, and it would make sense to reflect this in the title.**
>
> Good point! We have changed the title to: Evaluating Visual Counterfactual Explainers
>
> **How does Proposition 1 fit into your benchmarking narrative? Could the authors explain that more clearly?**
>
> The finding states that we should proceed with care when we assume the existence of a “ground truth” classifier, which is needed to evaluate the counterfactual explanation methods in the absence of a human evaluator. Our negative result formalizes the fact that a “ground truth” classifier is ill defined when the generator is not reversible, which is often the case with images. This also justifies the use of a synthetic image generator in our benchmark to have better control over reversibility.  We have emphasized this in Section 3.3.
>
> **In section 3.4 you say that “the goal of counterfactual generation methods is to find all the attributes that make a classifier behave differently from a causal classifier”. Since in practice we never usually know the causal data generating process, I am wondering whether this is really the goal of counterfactual generation methods. How would one evaluate this the achievement of this goal in practice?**
>
> Good point! In practice, this goal should be evaluated by humans. For instance, given a dog detector, a human knows that changing the background of the image should not affect the output of the model. In this case, the human knows that dog presence is a causal attribute for the classifier while the background is an independent or correlated variable. In the absence of humans (e.g. an automated benchmark), we resort to a well-defined data generation process where we have access to this information (correlated, causal, and independent variables) that we would not have in practice. This enabled us to perform an automated and fair comparison of multiple explainers. We have rewritten Section 3.4 to simplify the notation and emphasize on this point. We encourage the reviewer to take a look at the revised Section 3.4 along with Figure 1(c).

---

> > ### Comment · Reviewer_pgaK · 2022-11-22
> > **Thank you for your response.**
> >
> > Thank you for your response. Based on the presented experimental evaluation (and measure) on only one synthetic dataset, the generalization of your approach remains questionable. I think this is by far the most critical aspect of the work. Therefore, I am going to keep my score.

---

> > > ### Author Response · Authors · 2022-11-22
> > > **Reponse to reviewer pgaK**
> > >
> > > Thanks for taking the time to respond.
> > >
> > > Note that we evaluate our metric on 4 different dataset configurations, despite all of them being based on Synbols.
> > >
> > > We would really like to emphasize that **evaluation on non-synthetic settings is simply not possible since the true data generating process of a non-synthetic dataset is unknown and therefore we wouldn't have access to the causal factors**. Thus, the evaluation setup proposed in previous works is not only based around metrics that are easy to cheat  but it is also incomplete since it is conducted on datasets in which the true data generating process is unkown (CelebA), resulting in an evaluation setting in which we must rely on qualitative results to evaluate explainers. We make this point *several times* across the paper (see Section 1, Section 2, Section 3.1 or Section 3.4) and we discuss it even further in Section 5.
> > >
> > > Our work presents the **first principled evaluation of visual counterfactual explainers**. **This** is the most critical aspect of our work. We hope to have gotten this point across, let us know if you have any further concerns.

---

> ### Author Response · Authors · 2022-11-12
> **Reponse to reviewer pgaK (1/2)**
>
> Thanks for taking the time to review the paper and for your suggestions. Please kindly findly find our answers below
>
> **The generative model and specifically the dependencies represented in the model (Fig. 1 (a)) are not well-motivated.**
>
> Thanks for pointing this out. The graphical model expresses a common data generating process where an image is produced by some variables but only some of these variables have an effect on the label. For example, a picture in a dataset of dog breeds is described by the appearance of the dog, the background of the picture, the position of the dog in the picture, etc. However, only the dog has an influence on the label (z_causal -> x,y). At the same time, small dog breeds appear more often indoors. Therefore, the label (the dog breed) will be correlated with the correlated background variables  (y->z_corr). On the other hand, the exact position of the dog in the picture is completely independent from the breed (z_independent -> x but not y). Thus, an explanation showing that moving the dog in the picture changes the prediction of a classifier indicates an unexpected behavior. Likewise, an explanation showing that perturbing the background changes the prediction of a classifier indicates that the classifier has learned a spurious correlation. This motivates considering z_causal, z_corr, and z_independent in the data generating process. We have updated Section 3.1 with this motivation.
>
> **(a) the authors use only one (black-and white) synthetic dataset**
>
> We believe that simplicity is a strength of our benchmark. It allows us to pinpoint what exactly are the attributes that result in a concrete model behavior. We have updated the text to reinforce this point.
>
> **(b) the different evaluation choices are not well motivated**
>
> The main goal of  all design choices of the proposed evaluation is to guarantee fairness of comparison. For that, we had to make the following choices:
> 1. All explainers should be compared in the same attribute space.
> 2. Explainers must not be able to “cheat” to achieve a high score:
>     - Repeating the same explanation many times. This motivates measuring diversity.
>     - Providing uninformative explanations. This motivates using an “oracle” or causal classifier to discover unexpected predictions.
>     - Using random noise or unrelated data to explain a given image. This motivates the proximity and orthogonality constraints.
>
> We have updated the text to clarify the motivation of our evaluation choices.
>
> We have also clarified the rationale for the main aspects of our evaluation setup in the introduction and added a section in the Appendix discussing it. Further, we have also modified Section 3.4 simplifying the notation and adding examples to illustrate the rationale behind the properties we evaluate in explainers. We encourage the reviewer to take a look at the revised version of Section 3.4 along with Figure 1(c).
>
> **(c) it is not clear which results generalize to other settings /Do the results generalize to other datasets or other similar generative processes?**
>
> Good point, however the problem is that evaluation is not yet possible in other settings! For instance, the true data generating process of CelebA is unknown. Thus, our work presents **the first principled evaluation of counterfactual explanation methods.** We believe this is critical for the progress of this field. We have further emphasized this point in Section 5 and Section 3.4.
>
> **(d) the evaluated CE methods behave similarly according to the proposed measure (Eq. 7), so either the authors evaluate very similar methods, or the proposed measures are non-discriminative.Are the proposed measures discriminative and meaningful for evaluation (given that there is little variance in the results)?**
>
> Exactly! Our work shows that, since most methods were originally evaluated on CelebA using metrics that are easy to game (see Sections 1 & 2 ), there has been little real improvement in recent years. This is an important conclusion of our work and we have added it in the discussion session.

---

> ### Author Response · Authors · 2022-11-18
> **Follow Up**
>
> Dear reviewer,
> We would like to make sure that all your concerns were addressed for this work. Please let us know if there is anything you would like to discuss further.

---

### Official Review · Reviewer_e2UL · 2022-11-02

**Confidence:** 2
**Correctness:** 2
**Technical Novelty And Significance:** 2
**Empirical Novelty And Significance:** 2
**Recommendation:** 5

**Clarity, Quality, Novelty And Reproducibility:**

- I didn't find the exposition particularly clear. I think many terms were quite loaded and terminology somewhat confusing. I think it could benefit from a worked clear example before experimentation to communicate the main themes better to the reader.
- The innovations here seem quite interesting. I think there is value in my understanding of the process described, but i felt that too many of the definitions used were un-justified. This is not to say they are unjustifiable, but I think the authors should have spent some convincing the readers that these are properties that are worth having.
- The authors linked to the code repository for their empirical results. Although the code doesn't have a 'main' file that will reproduce the results in the paper, the code looks very clear, well documented and extensive. If I didn't miss the main file in the repo (it looks like benchmark covers much of the experiments here), I would encourage the authors to add it whenever possible.
- The main contributions seem to be the TF, SCE and orthogonal components. These seem quite interesting and useful contributions. Although I ask for a little more clarity and justification in some of these, I think they are quite reasonable and

**Strength And Weaknesses:**

Strengths:
- Highlights some weaknesses of counterfactual evaluation, and proposes interesting solutions to important gaps in the literature.
- I found the conclusion that different counterfactual explaners in the literature perform similarly to be particularly interesting. I would welcome additional experiments and datasets to confirm this more broadly.
- The contribution of the paper offering a more comprehensive definition of good counterfactual is very appealing

Weaknesses
- Several aspects (e.g. FN, FNF, TF, etc) were defined, but I didn't feel the rationale behind why these are useful and relevant properties for counterfactual explanations was sufficiently covered.
- I would have welcomed some additional discussion about $z_causal$. I understand how you have access to this in the experiments (synbols), but in a more general context, how hard is it to get acces to these traits? Is there room for noisy estimates on these?
- Similar to the above, I would have welcomed discussion about the dependency on oracles for the conclusions.
- I wasn't convinced that the results are generally applicable, particularly given the narrow scope of the experiments. The experimental section focuses on one dataset that is rather tightly controlled. This helps us understand the dynamics of the problem very well, but I don't think the discussion allows me to see this as more than experiments on toy contexts. This would not be too problematic, but I feel like the title and conclusions claim a lot based on the experiments. I know that in the appendix some real-world considerations are discussed, but to me these are far too important to leave in the apendix.
- For me, there are missing links between the results as presented here and the general problem. For me, a section or paragraph illustrating highlighting opportunities for readers to deploy the lessons of this paper to new environments would have been useful. The results are empirical on a well controlled environment. How would this be taken forward?

Misc. & smaller points
- I found the notation difficult to follow, which hindered me understanding the major contributions. I'm slightly unfamiliar with the literature, so this may be usual convention.
- The legend on Fig 3 is far too small

**Summary Of The Paper:**

This paper aims at establishing a comprehensive definition of counterfactuals explanations. After reviewing related work, it establishes the foundations for establishing their process: data generation, counterfactual generation, classifiers, and evaluation. Their approach is compared to a number of baselines on one particular dataset which they can control explicitly, and a number of interesting conclusions are drawn from the experiments.

**Summary Of The Review:**

This paper has made some interesting contributions to the literature. I found it difficult to understand whether the conclusions are can be applied broadly to explainability in general, or whether it's particular to the experimental context explored. I think that the main contributions are interesting and useful too, but that some discussion around real-world and practical deployment was lacking.

---

> ### Author Response · Authors · 2022-11-12
> **Response to reviewer e2UL (2/2)**
>
> **I wasn't convinced that the results are generally applicable, particularly given the narrow scope of the experiments. The experimental section focuses on one dataset that is rather tightly controlled. This helps us understand the dynamics of the problem very well, but I don't think the discussion allows me to see this as more than experiments on toy contexts. This would not be too problematic, but I feel like the title and conclusions claim a lot based on the experiments. I know that in the appendix some real-world considerations are discussed, but to me these are far too important to leave in the apendix.**
>
> Thanks for this comment! We have moved the real-world considerations to the discussion section (Section 5). We would like to remark that our evaluation would not be possible without the use of a synthetic dataset since,  without further assumptions about the data, it is not possible to access their true latent variables. Thus, knowing the true data generating process allows us to know whether a model prediction should change after perturbing a latent variable and identify unexpected modes of behavior (we have added this insight in Section 3.4). Please note that if an explainer fails to provide a set of useful and diverse explanations for our synthetic dataset it is unlikely that it is able to do so for real datasets. We have moved this consideration into the main text (Section 5).
>
> **For me, there are missing links between the results as presented here and the general problem. For me, a section or paragraph illustrating highlighting opportunities for readers to deploy the lessons of this paper to new environments would have been useful. The results are empirical on a well controlled environment. How would this be taken forward?**
>
> Good point. An important benefit of our benchmark is that it provides all the tools to test counterfactual explanation methods in an efficient way since images are small and there is no need to train a model for latent decomposition because we already provide a common one for all explainers, resulting in a fair evaluation setup. Thus, the benchmark could be used to efficiently test new explainability methods before trying them on larger datasets for which their latent attributes are unknown. We hope our benchmark seeds up progress in counterfactual explainability. Further, note that the complete evaluation is not yet possible in other settings! For instance, the true data generating process of CelebA is unknown. Thus, our work presents **the first principled evaluation** of counterfactual explanation methods. We believe this is critical for the progress of this field. We have added this in the text.
>
> (1) **I found the notation difficult to follow, which hindered me understanding the major contributions. I'm slightly unfamiliar with the literature, so this may be usual convention.**
>
> (2) **I didn't find the exposition particularly clear. I think many terms were quite loaded and terminology somewhat confusing. I think it could benefit from a worked clear example before experimentation to communicate the main themes better to the reader.**
>
> (3) **The innovations here seem quite interesting. I think there is value in my understanding of the process described, but i felt that too many of the definitions used were un-justified. This is not to say they are unjustifiable, but I think the authors should have spent some convincing the readers that these are properties that are worth having.**
>
> To address all of these concerns we have rewritten Section 3.4, simplifying the notation and providing examples as to why causal knowledge is important when evaluating the usefulness of counterfactuals. We encourage the reviewer to take a look at the revised Section 3.4 along with figure 1(c).
>
> **The authors linked to the code repository for their empirical results. Although the code doesn't have a 'main' file that will reproduce the results in the paper, the code looks very clear, well documented and extensive. If I didn't miss the main file in the repo (it looks like benchmark covers much of the experiments here), I would encourage the authors to add it whenever possible.**
>
> Thanks for checking out the repo!  We have added a run_benchmark.py file in the repository along with instructions on how to reproduce the experiments in the paper in the README as per your suggestion.

---

> ### Author Response · Authors · 2022-11-12
> **Respose to reviewer e2UL (1/2)**
>
> Thanks for your feedback and suggestions! Please kindly find our answers below.
>
> **Several aspects (e.g. FN, FNF, TF, etc) were defined, but I didn't feel the rationale behind why these are useful and relevant properties for counterfactual explanations was sufficiently covered.**
>
> Thanks for pointing this out.  We have rewritten Section 3.4, simplifying the notation and adding examples to illustrate the rationale behind the properties we evaluate in explainers. Concretely we have added the following example:
> > “...having knowledge of the causal factors (access to $h_{\text{causal}}$)  allows us to evaluate counterfactuals explanations in a new way. Consider the following scenario: Given a dog classifier and an image of a dog, a counterfactual example that removes the dog from the image without altering the classifier's prediction (CF) will almost certainly provide valuable insight about the model's behavior. The same can be said about a counterfactual example that only changes the background of the image in a way that alters the classifier's prediction (NCF)”.
>
> We encourage the reviewer to take a look at the revised Section 3.4 along with figure 1(c)
>
> **I would have welcomed some additional discussion about zcausal. I understand how you have access to this in the experiments (synbols), but in a more general context, how hard is it to get access to these traits? Is there room for noisy estimates on these?**
>
> Good point. In the general cases, the true latent variables of the data generating process are not identifiable [1]. However, using further assumptions it is possible to identify the latent variables [2]. Moreover, in a temporal setup, it is possible to identify which of these latent variables are the causal ones [3]. Finally, in a multi-task setup where distribution shift occurs, it is possible to identify which variables are robust to distributions shift and hence, likely to be the causal ones. Thus, although it would be possible to approximate the true latent factors in some cases, it would require making additional assumptions about the data. We have included this reflection in the Appendix.
>
> [1] Locatello, Francesco, et al. "Challenging common assumptions in the unsupervised learning of disentangled representations." international conference on machine learning. PMLR, 2019.
>
> [2] Khemakhem, Ilyes, et al. "Variational autoencoders and nonlinear ica: A unifying framework." International Conference on Artificial Intelligence and Statistics. PMLR, 2020.
>
> [3] Lachapelle, Sébastien, et al. "Disentanglement via mechanism sparsity regularization: A new principle for nonlinear ICA." Conference on Causal Learning and Reasoning. PMLR, 2022.
>
>
> **Similar to the above, I would have welcomed discussion about the dependency on oracles for the conclusions.**
>
> The oracle, which could be a human, is required to contrast the predictions of a model with the optimal prediction and spot unexpected behaviors. Without an oracle, we could not assess whether a model is working as intended. We have included this reasoning in the discussion (Section 5).

---

> ### Author Response · Authors · 2022-11-18
> **Follow Up**
>
> Dear reviewer,
> We would like to make sure that all your concerns were addressed for this work. Please let us know if there is anything you would like to discuss further.

---

> > ### Comment · Reviewer_e2UL · 2022-11-28
> > **Lingering uncertainty about experiments, but open to change of heart**
> >
> > Thanks to the authors for responding to the comments in my review. I appreciate the time and consideration that went into addressing my concerns. I think most of them are now reduced, but I do find myself questioning whether I'm convinced that the experimental section is sufficient for us to draw general conclusions. I work in a context where simulated data is a critical mechanism for understanding and demonstrating the true value of the model under investigation. I also get frustrated when reviewers are dismissive of the claims of my papers that are demonstrated in toy contexts. So please understand that I am not discounting the value of your quandry, since indeed I'm quite empathetic to it. So, i don't think I expect new experiments, but I think I could strengthen my scoring of the paper if there was a middle ground relayed, something akin to a practical guid on how to deploy such a model on a real human-in-the-loop situation in the appendix. I think I could also be convinced if a reviewer who is more familiar with the field would advocate for it.

---

> > > ### Author Response · Authors · 2022-12-06
> > > **Changing Reviewer e2UL's heart**
> > >
> > > Thanks for responding to our comments and for your understanding. We appreciate it.
> > >
> > >
> > > We would like to emphazise that in this case, **evaluating on non-synthetic settings is simply not possible since the true data generating process of a non-synthetic dataset is unknown and therefore we wouldn't have access to the causal factors**.  We modified the text in the revision to further emphasize this point across the paper (see Section 1, Section 2, Section 3.1 or Section 3.4 and Section 5).  Nevertheless, it is possible to adapt our metric to take only into account an orthogonal and complement set of estimator flips $\text{EF}$ (Eq. 3) which do not require causal knowledge. However, any evaluation schema that does not include causal information would be incomplete. We added this insight in the discussion section of the paper (Section 5) in the revised version.
> > >
> > > Before our work, explainers were evaluated on metrics that were easy to cheat and without access to the true causal factors. Our work presents **the first principled evaluation of visual counterfactual explainers**. Let us know if you have any further concerns.

---

### Official Review · Reviewer_WjLU · 2022-11-02

**Confidence:** 4
**Correctness:** 2
**Technical Novelty And Significance:** 2
**Empirical Novelty And Significance:** 2
**Recommendation:** 3

**Clarity, Quality, Novelty And Reproducibility:**

Clarity: The paper is difficult to understand and some of the assumptions are not clear (see the detailed review above).

Quality: The setting considered in the paper is very restrictive and the effectiveness of the proposed solutions on more general settings is not very clear.

Novelty: The paper largely ignores existing work on causality when generating counterfactuals. Similarly, there is existing work on generating "ground truth" explanations which should be added to the discussion (see main review).

Reproducibility: The code will be open sourced on publication.

**Strength And Weaknesses:**

### Strengths
Lack of reliable evaluation methods for counterfactuals (and explanations in general) is indeed an important problem.

### Weaknesses
There are three main weaknesses in the paper: 1) The setting in the paper is overly restrictive and it is not clear how useful the proposed metric is in the real world. 2) Some of the assumptions taken in the paper are not backed up with supportive arguments and seem at odds with what one would expect from explainers. 3) The clarity of writing can be significantly improved. For these reasons, the paper is not ready yet. Please see detailed comments and questions below:

1. First of all, the focus is mainly on the synthetic synbols dataset where the real latent attributes are available and the data generation process can be controlled. On the other hand, learning these two entities constitutes one of the main challenges in learning. While this does not mean that simplified synthetic settings should not be studied, the paper does not provide any indication on how the takeaways from here would generalize to real datasets.

2. Some of the basis assumptions are not clear. In section 3.2, why is it more useful to perturb "non-causal" attributes? The model output could be sensitive to causal or non-causal attributes. A faithful explainer should reflect the true behavior of the model. Similarly, the concept of “trivial” explanation is not very clear. Assuming that the “oracle” here refers to a human, there could be perturbations that do not change the human’s prediction, but do change the prediction of the model. For instance, the model might change its prediction based on the brightness of the image (while the human will not). That could be deemed a valid counterfactual explanation since it correctly reflects the behavior of the model.

3. The paper correctly points out issues with existing diversity metrics in the literature. However, I am not sure if using a single “principled metric to compare counterfactual explanation methods” is the answer. One could always use different metrics to study different aspects of explanation quality. Or does the paper mean to make the point that a single metric can evaluate all relevant aspects of an explanation?

4. There has already been some work on considering causality when generating counterfactual explanations (https://arxiv.org/pdf/2002.06278.pdf). It would be good to contrast the approach in Sections 3.3 and 3.4 with the existing work.

5. It is fine to focus on generative models for counterfactual explanations. However, note that there exist other methods for generating counterfactuals that do not rely on the latent attributes see for instance (https://arxiv.org/abs/1910.08485) and (https://arxiv.org/pdf/1912.09405.pdf).

6. The exposition could be improved at several places. For instance, what is "the oracle". What is $\hat{h}$ and how is it different from $\hat{f}$. Since the paper introduces quite a bit of terminology, I would suggest putting all the terms together in a single table so that the reader can easily look it up, and clearly defining the input / outputs of different functions.

7. It would help to differentiate from the work of Zhou et al (https://arxiv.org/abs/2104.14403) which has a very similar goal: adding “ground truth“ perturbations to the images that the explainers should pick up on.

**Summary Of The Paper:**

The paper proposes an evaluation strategy for the counterfactual explanations of images with focus on generative models that operate in the latent space. The paper points out flaws with existing strategies: the diversity criterion can be fooled by making irrelevant changes and the real latent attributes could be misaligned with the learnt latent attributes. The proposed metric works by disentangling the causal vs. non-causal relationships in the latent space. Experiments are reported with several methods.

**Summary Of The Review:**

The paper does not meet the bar for ICLR. The solution is applicable to only very restrictive settings and the main assumptions are not backed up with arguments. The paper is also quite hard to follow at several places.

---

> ### Author Response · Authors · 2022-11-11
> **Respose to reviewer WjLU (2/2)**
>
> **Assuming that the “oracle” here refers to a human, there could be perturbations that do not change the human’s prediction, but do change the prediction of the model. For instance, the model might change its prediction based on the brightness of the image (while the human will not). That could be deemed a valid counterfactual explanation since it correctly reflects the behavior of the model.**
>
> Exactly! Our benchmark is the first to consider these types of explanations :)
> This is summarized in the gray regions in Figure1(c). Note that this  is precisely the kind of explanation that can only be evaluated by having full control over the entirety of the data generation process along with the causal and non-causal attributes, which is what our benchmark provides. We have rewritten Section 3.4 simplifying the notation and adding more examples to make this more clear. We encourage the reviewer to take a look at the revised Section 3.4 along with Figure 1(c) .
>
>
> **The paper correctly points out issues with existing diversity metrics in the literature. However, I am not sure if using a single “principled metric to compare counterfactual explanation methods” is the answer. One could always use different metrics to study different aspects of explanation quality. Or does the paper mean to make the point that a single metric can evaluate all relevant aspects of an explanation?**
>
> Good point! Our metric highlights important properties of an explainer and its design is aligned with usefulness. However it does not depict all the properties of an explainer such as fragility, speed, etc. We have updated the Limitations section with this point.
>
> **There has already been some work on considering causality when generating counterfactual
> explanations (https://arxiv.org/pdf/2002.06278.pdf). It would be good to contrast the approach in Sections 3.3 and 3.4 with the existing work.**
>
> This paper indeed looks interesting, however note that we focus on the image domain where access to the causal generating process is not possible most of the time. To the best of our knowledge our work is the first one to consider causal factors when evaluating counterfactuals in the image domain providing a fair and principled evaluation scheme.
>
> **It is fine to focus on generative models for counterfactual explanations. However, note that there exist other methods for generating counterfactuals that do not rely on the latent attributes see for instance (https://arxiv.org/abs/1910.08485) and (https://arxiv.org/pdf/1912.09405.pdf).**
>
> Thanks for the references! We have added them in the related work section. Note that both these works are attribution methods, which have some limitations when compared to counterfactual generation methods as we mentioned in the related work section:
> >“Another explainability technique is visualizing the factors that influenced a model's decision through heatmaps. Heatmaps are useful to understand which objects present in the image have contributed to a classification. However, heatmaps do not show how areas of the image should be changed and they cannot explain factors that are not spatially localized (e.g., size, color, brightness, etc).”
>
> Also note that our method relies on the latent space since counterfactual examples found in the pixel space often resemble adversarial attacks (see Figure 4).
>
> **The exposition could be improved at several places. For instance, what is "the oracle". What is h^ and how is it different from f^**
>
> We agree that definitions of $\hat{h}$ and $\hat{f}$ are very similar, but they are explicitly defined inSection 3.2. Concretely $\hat{f}$ is a classifier that takes as input an image $x$ and $\hat{h}$ is equivalent to the classifier $\hat{f}$ when fed an image generated by generator $g(\textbf{z})$ so: $\hat{h}:= \hat{f}(g(\textbf{z}))$. Please see the notation table in the Appendix (Table 3). Oracle is a standard term in the literature and refers to an agent/system with knowledge of the causal factors that generated the data. Nevertheless, we have clarified this in the text as per your suggestion.
>
>
> **Since the paper introduces quite a bit of terminology, I would suggest putting all the terms together in a single table so that the reader can easily look it up, and clearly defining the input / outputs of different functions.**
>
> Please see the notation table in the Appendix (Table 3).
>
>
> **It would help to differentiate from the work of Zhou et al (https://arxiv.org/abs/2104.14403) which has a very similar goal: adding “ground truth“ perturbations to the images that the explainers should pick up on.**
>
> Thanks for the reference! We have included it in the related work section. The main difference between this work and ours is that our work focuses on counterfactual generation methods, while this work focuses on attribution methods, which have limitations when compared to counterfactual generation methods (see above).

---

> ### Author Response · Authors · 2022-11-11
> **Respose to reviewer WjLU (1/2)**
>
> Thanks for your positive feedback and detailed review! Please kindly find our reply below.
>
> **The setting in the paper is overly restrictive and it is not clear how useful the proposed metric is in the real world. Some of the assumptions taken in the paper are not backed up with supportive arguments and seem at odds with what one would expect from explainers.**
>
> We tried to impose the minimal possible number of restrictions to achieve fairness of comparison:
> 1. All explainers should be compared in the same attribute space
> 2. Explainers must not be able to “cheat” to achieve a high score:
>     - Repeating the same explanation many times
>     - Providing uninformative explanations: An explanation is informative if it deviates from the expected value. For existing images, we know the expected value of the label will be the true label. However, for new images we do not know the true label. Thus, we are required to compare with some sort of oracle or ground truth classifier that tells us the true label. This is impossible for real datasets such as CelebA, since attributes are subjective (e.g., attractiveness), and they do not fully describe the images (e.g., the background color is not specified).
>     - Using random noise or unrelated data to explain a given image. This motivates the proximity constraint, which is widespread in the literature.
>
> We believe that this is a minimal set of well-fundamented restrictions needed for fair comparison and we have updated the text to make sure that this is clear.
>
> **The focus is mainly on the synthetic synbols dataset where the real latent attributes are available and the data generation process can be controlled. On the other hand, learning these two entities constitutes one of the main challenges in learning. While this does not mean that simplified synthetic settings should not be studied, the paper does not provide any indication on how the takeaways from here would generalize to real datasets.**
>
> Our synthetic benchmark is meant to evaluate properties of an explainer that could not be evaluated on real world data in a principled way. We recommend the user to also evaluate explainers using real world data. We have moved this insight from the Limitations section (A.4) to the Discussion section (Section 5) along with some insight as to how our metric can be adapted for real world data:
> >“...when working in real world scenarios it is possible to adapt our metric to take only into account an orthogonal and complement set of estimator flips (EF) which do not require knowledge of the causal attributes. However, any evaluation schema that does not include causal information would be incomplete.  Further, note that if an explainer fails to provide a set of useful and diverse explanations for our synthetic dataset it is very unlikely that it is able to do so for real datasets. Nevertheless, we recommend users to also evaluate explainers using real world data”.
>
> **Some of the basis assumptions are not clear. In section 3.2, why is it more useful to perturb "non-causal" attributes? The model output could be sensitive to causal or non-causal attributes. A faithful explainer should reflect the true behavior of the model. Similarly, the concept of “trivial” explanation is not very clear.**
>
> We know the expected prediction when changing causal variables (e.g., for a dog classifier, we expect that removing dogs from an image would change its prediction because dog presence is the cause of the “dog” label), explanations of this type were named as “trivial” by Rodriguez et al. Thus, to find an informative explanation we should look for deviations from the expected behavior (e.g., dogs are removed but the model being explained does not change its prediction, or the background color is changed and the model being explained does change its prediction) these cases are summarized in Figure 1c.  and we have added this example in Section 3.4.
> We also provide an example of a “trivial” explanation as mentioned in Section  3.3:
> >“Counterfactual explanation methods tend to produce trivial explanations by perturbing the attribute being classified from the input”.

---

> ### Author Response · Authors · 2022-11-18
> **Follow Up**
>
> Dear reviewer,
> We would like to make sure that all your concerns were addressed for this work. Please let us know if there is anything you would like to discuss further.

---

### Official Review · Reviewer_jg6E · 2022-11-03

**Confidence:** 3
**Correctness:** 3
**Technical Novelty And Significance:** 4
**Empirical Novelty And Significance:** 4
**Recommendation:** 6

**Clarity, Quality, Novelty And Reproducibility:**

The work is quite clear and novel. I did not attempt to use the code they say they posted.

**Strength And Weaknesses:**

Strengths: The authors provide lots of helpful exposition on why they proposed the method in the way that they did, and are clear about their stance on what counterfactual explanations should be. The insights they draw at the end are interesting; for instance, "explainers fail to consistently find more than one [attribute to alter to generate interesting counterfactuals]."

Weaknesses: The premise of the paper posits VERY strong assumptions on the *purpose* of counterfactual explanations: to be diverse, and to illustrate tensions with an optimal causal model. Though I don't deny these are *nice* qualities, I think their introduction could be clarified to assert *why* those properties are desirable (and why they did not consider other qualities). It may seem self explanatory, but really break it down: Why is it useful to know when a model is not acting "causally"??

Further, although their conclusions from applying their benchmark to explanation methods are interesting and believable, they seem to involve a bit of speculation, sometimes providing "because" explanations for the results they see that don't seem to have been directly experimentally validated. Sometimes, I felt the authors went off on tangents not totally necessary for the paper; for instance I am not sure why Proposition 1 is important.

**Summary Of The Paper:**

The authors propose a novel way to compare counterfactual explanation methods, which "explain" why a classifier made a certain prediction by highlighting nearby points in the domain with a different outcome. Their proposal involves a synthetic dataset in which the latent causal drivers of the outcome are "known," and greedily constructing a set of explanations which are "orthogonal" or "complementary" to each other. This method is built from the premise that "The goal for counterfactual generation methods is to find all the attributes that make a classifier behave differently from a causal classifier" and "a good explainer should return a ... diverse set of explanations." They use their method to benchmark existing methods for generating counterfactuals.

**Summary Of The Review:**

I believe this work is interesting and worthy of publication, however I think it could use some polishing, and perhaps a redistribution of emphasis (i.e. providing more results in the body of the paper rather than delegating to the appendix and briefly summarizing them at the end)

---

> ### Author Response · Authors · 2022-11-11
> **Response to reviewer jg6E**
>
> Thank you for your feedback! Please kindly find our replies below.
>
>
> **The premise of the paper posits VERY strong assumptions on the purpose of counterfactual explanations: to be diverse, and to illustrate tensions with an optimal causal model. Though I don't deny these are nice qualities, I think their introduction could be clarified to assert why those properties are desirable (and why they did not consider other qualities). It may seem self explanatory, but really break it down: Why is it useful to know when a model is not acting "causally"??**
>
> Thanks for asking, this is a very important point. We aim to improve the “informativeness” of explanation sets. In terms of information theory, unexpected events tend to be more informative. In this sense, repeating the same explanation multiple times provides little information and increasing diversity increases the amount of information. In addition, the amount of information of an event is usually measured with respect to some underlying probability distribution. If we know that a classifier is trained to detect a certain object, we expect the classifier to learn the conditional probability distribution (label given input) of the training set. Thus, deviations from this distribution are unexpected and informative. However, we do not know the real label for new samples and thus, we need to resort to some form of oracle or ground truth classifier in order to compare it with the model being explained. This ground truth classifier must have access to the real data generating process in order to infer the correct class from images, and the data generating process of image-label distributions is typically a causal process governed by the laws of physics. That is why we refer to this classifier as “causal classifier”. We have clarified this in the introduction and added a section in the Appendix discussing it.
>
> **Further, although their conclusions from applying their benchmark to explanation methods are interesting and believable, they seem to involve a bit of speculation, sometimes providing "because" explanations for the results they see that don't seem to have been directly experimentally validated.**
>
> Thanks for pointing this out. We have reviewed Section 4.2  and removed all the claims and hypotheses that are not experimentally validated, leaving only those that are well supported. Concretely we changed the wording to avoid speculation and removed the paragraph: **Counterfactual explainers are far from optimality**.
>
> **Sometimes, I felt the authors went off on tangents not totally necessary for the paper; for instance I am not sure why Proposition 1 is important.**
>
> Thanks for this comment! The finding states that we should proceed with care when we assume the existence of a “ground truth” classifier, which is needed to evaluate the counterfactual explanation methods in the absence of a human evaluator. Our negative result formalizes the fact that a “ground truth” classifier is ill defined when the generator is not reversible, which is often the case with images. More importantly, this also justifies the use of a synthetic image generator in our benchmark to have better control over reversibility.  We have emphasized this in Section 3.3.

---

> ### Author Response · Authors · 2022-11-18
> **Follow Up**
>
> Dear reviewer,
> We would like to make sure that all your concerns were addressed for this work. Please let us know if there is anything you would like to discuss further.

---

### Author Response · Authors · 2022-11-11
**Thanks for your valuable feedback**

We would like to thank the reviewers for their positive and thoughtful feedback, the changes they proposed have helped us improve the clarity of the paper. We are encouraged that they found our work *"Highlights some weaknesses of counterfactual evaluation, and proposes interesting solutions to important gaps in the literature"* (e2UL), and that *“the proposed idea for deriving evaluation measures for counterfactual explanations is sensible.”* (pgaK).  They also acknowledge that the *“lack of reliable evaluation methods for counterfactuals (and explanations in general) is indeed an important problem”* (WjLU) and that *“the authors provide lots of helpful exposition on why they proposed the method in the way that they did, and are clear about their stance on what counterfactual explanations should be”* ( jg6E).

We have modified the manuscript to address their concerns, improving the clarity and emphasizing the following point.

The main goal of all design choices of the proposed evaluation is to guarantee fairness of comparison. For that, we had to make the following choices:

1. All explainers should be compared in the same attribute space.
2. Explainers must not be able to “cheat” to achieve a high score:
    - Repeating the same explanation many times. This motivates measuring diversity.
    - Providing uninformative explanations. This motivates using an “oracle” or causal classifier to discover unexpected predictions.
    - Using random noise or unrelated data to explain a given image. This motivates the proximity and orthogonality constraints.

We have uploaded a revision of the paper *with the changes in blue* so that they are easier to spot. We encourage the reviewers to take a look at it.

Please kindly find our individual replies below.

---

### Decision · Program_Chairs · 2023-01-20

**Decision:**

Reject

**Justification For Why Not Higher Score:**

Narrow problem setting. Insufficient motivation of assumptions. Presentation issues.

**Justification For Why Not Lower Score:**

N/A

**Metareview: Summary, Strengths And Weaknesses:**

The paper proposes an evaluation strategy for the counterfactual explanations of images with focus on generative models that operate in the latent space. The reviewers did recognize some merit to the paper, however, they highlighted a large number of concerns. More prominently, the reviewers think that problem setting is overly restrictive, the motivation for some of the assumptions as well as the contextualization with related work is lacking, and the presentation can be improved. The response/discussion period did not persuade the reviewers to support acceptance.